# Phosphoproteomics reveals rewiring of the insulin signaling network and multi-nodal defects in insulin resistance

Daniel J. Fazakerley ●[1,2,10] ✉, Julian van Gerwen[1,10], Kristen C. Cooke[1], Xiaowen Duan[1], Elise J. Needham[1], Alexis Díaz-Vegas[1], Søren Madsen[1], Dougall M. Norris[2], Amber S. Shun-Shion ●[2], James R. Krycer[1,3,4], James G. Burchfield ●[1], Pengyi Yang ●[5,6], Mark R. Wade[7], Joseph T. Brozinick ●[7], David E. James ●[1,8] ✉ & Sean J. Humphrey ●[1,9] ✉

The failure of metabolic tissues to appropriately respond to insulin ("insulin resistance") is an early marker in the pathogenesis of type 2 diabetes. Protein phosphorylation is central to the adipocyte insulin response, but how adipocyte signaling networks are dysregulated upon insulin resistance is unknown. Here we employ phosphoproteomics to delineate insulin signal transduction in adipocyte cells and adipose tissue. Across a range of insults causing insulin resistance, we observe a marked rewiring of the insulin signaling network. This includes both attenuated insulin-responsive phosphorylation, and the emergence of phosphorylation uniquely insulin-regulated in insulin resistance. Identifying dysregulated phosphosites common to multiple insults reveals subnetworks containing non-canonical regulators of insulin action, such as MARK2/3, and causal drivers of insulin resistance. The presence of several bona fide GSK3 substrates among these phosphosites led us to establish a pipeline for identifying context-specific kinase substrates, revealing widespread dysregulation of GSK3 signaling. Pharmacological inhibition of GSK3 partially reverses insulin resistance in cells and tissue explants. These data highlight that insulin resistance is a multi-nodal signaling defect that includes dysregulated MARK2/3 and GSK3 activity.

Insulin resistance is a key defect preceding type 2 diabetes, cardiovascular disease, and other metabolic diseases[1]. Impaired insulin-stimulated translocation of the glucose transporter GLUT4 to the cell surface is one of the primary defects resulting from insulin resistance in muscle and adipose tissue, leading to decreased insulin-stimulated glucose uptake[1]. Adipose tissue insulin resistance precedes muscle insulin resistance during high-fat feeding[2,3], and adipose-specific abrogation of insulin action leads to systemic insulin resistance[4,5].

[1]Charles Perkins Centre, School of Life and Environmental Sciences, University of Sydney, Sydney, NSW 2006, Australia. [2]Metabolic Research Laboratories, Wellcome-Medical Research Council Institute of Metabolic Science, University of Cambridge, Cambridge CB2 0QQ, UK. [3]QIMR Berghofer Medical Research Institute, Brisbane, QL, Australia. [4]Faculty of Health, School of Biomedical Sciences, Queensland University of Technology, Brisbane, QL, Australia. [5]Charles Perkins Centre, School of Mathematics and Statistics, University of Sydney, Sydney, NSW 2006, Australia. [6]Computational Systems Biology Group, Children's Medical Research Institute, Faculty of Medicine and Health, University of Sydney, Westmead, NSW 2145, Australia. [7]Lilly Research Laboratories, Division of Eli Lilly and Company, Indianapolis, IN, USA. [8]Sydney Medical School, University of Sydney, Sydney 2006, Australia. [9]Murdoch Children's Research Institute, The Royal Children's Hospital, Melbourne, VIC 3052, Australia. [10]These authors contributed equally: Daniel J. Fazakerley, Julian van Gerwen. ✉e-mail: djf72@medschl.cam.ac.uk; david.james@sydney.edu.au; sean.humphrey@mcri.edu.au

Hence, there is considerable interest in studying adipose insulin action to understand how this tissue contributes to whole-body insulin resistance. Reversible protein phosphorylation is an essential post-translational modification mediating insulin action, making signal transduction a focal point for mechanistic understanding of this metabolic process[6].

Adipose insulin resistance has been linked to external stressors including chronic inflammation[7], hyperinsulinemia[8], glucocorticoids[9], and lipotoxicity[10,11], as well as associated internal stressors such as mitochondrial dysfunction[12], oxidative stress[13], and endoplasmic reticulum stress[14]. These stressors have been suggested to impair early steps in the insulin signaling cascade, thereby attenuating insulin-stimulated GLUT4 translocation. Indeed, several protein kinases activated by these insults, including mTORC1, S6K, Jnk and PKC, have been reported to phosphorylate and inhibit the insulin receptor or its scaffold protein IRS (reviewed in[15]). However, we and others have recently called this hypothesis into question[1,5,16], and the causal relationship between changes in early insulin signaling and the downstream actions of insulin remains unclear, especially in the context of insulin resistance. There is therefore an urgent need for global unbiased analyses of signaling in insulin-resistant cells to pinpoint potential alterations contributing to insulin resistance.

Mass spectrometry (MS)-based phosphoproteomics now makes it possible to study insulin signaling on a global scale[17–19]. Here, we employed recent advances in the throughput and sensitivity of these technologies[20,21] to analyze acute signaling responses to insulin in the context of adipose insulin resistance. We initially focused on the highly insulin-responsive 3T3-L1 adipocyte cell line allowing us to study signaling defects in isolation from the complex milieu of whole organisms, and subsequently validated key findings in mouse adipose tissue. In particular, we exposed cells to five insults spanning a broad range of factors known to contribute to adipocyte insulin resistance, reasoning that signaling changes induced by multiple insults would be more likely to contribute to the etiology of insulin resistance.

Our global phosphoproteomic analyses reveal that insulin resistance is accompanied by extensive rewiring of the insulin signaling network, which is largely exclusive of canonically studied insulin signaling nodes. A diverse subnetwork of kinases, proteins, and pathways were dysregulated across multiple models, prompting us to further investigate MARK2 and MARK3 as regulators of insulin action. Using a small molecule inhibitor together with phosphoproteomics and motif analysis we generated a resource of adipocyte-specific substrates for the kinase GSK3. This approach facilitated the identification of defective GSK3 signalling across all cell models, and in insulin-resistant adipose tissue. Studies in cells and tissues using multiple GSK3 inhibitors established that acute GSK3 inhibition partially restored insulin sensitivity, supporting these findings. Thus, our data provide a quantitative atlas delineating the signaling dysregulation in adipocytes during insulin resistance and can be navigated at www.adipocyteatlas.org.

## Results

### Establishing proteomic and phosphoproteomic models of adipocyte insulin resistance

We selected five distinct models of insulin resistance in 3T3-L1 adipocytes, enabling us to pinpoint molecular rearrangements specific to insulin resistance rather than unrelated effects of the insults that cause it. These included three well-characterized models mimicking systemic insults reported to induce insulin resistance in humans and animals: hyperinsulinemia (chronic insulin treatment, **CI**), glucocorticoids (dexamethasone treatment, **DEX**) and inflammation (TNFα treatment, **TNF**) (Fig. 1a)[3]. Mitochondrial oxidants have recently been implicated in insulin resistance[3,22,23], and we therefore included two models comprising the acute (2h) production of mitochondrial oxidants from different sources (mitochondria-targeted paraquat, **MPQ**, antimycin A,

**AA**) (Fig. 1a)[22,23]. All five models conferred impaired uptake of radiolabeled 2-deoxyglucose (2DG) into adipocyte cells following an acute (10 min) insulin treatment, confirming that cells were insulin resistant (Fig. 1b).

We assessed changes in protein expression and insulin signaling responses in control and insulin-resistant adipocytes by mass spectrometry-based proteomic and phosphoproteomic analyses (Fig. 1c). Together we quantified 7564 proteins across all models (Fig. 1d), as well as 39,846 phosphopeptides corresponding to 29,311 phosphorylation sites on 3791 proteins (Fig. 1e). Our proteomic and phosphoproteomic data were very high quality, revealed by Pearson's correlation, principal component analysis, and hierarchical clustering, as samples from the same model and insulin-stimulation status were highly correlated (average Pearson's correlation; proteome r = 0.96, phosphoproteome r = 0.89) and clustered together (Supplementary Fig. 1a–f). We also observed no major differences in the number of quantified phosphopeptides and proteins between conditions (Supplementary Fig. 1g, h). Together, this demonstrates that our MS data are both accurate and reproducible. While unstimulated and insulin-stimulated samples from the same model clustered distinctly when considering the phosphoproteome data (Supplementary Fig. 1a, c), they clustered together in the proteome data (Supplementary Fig. 1b, d), indicating that the proteome was unchanged in the short 10 min insulin treatment, as expected. Accordingly, we detected no statistically significant changes to the proteome upon the short insulin treatment across all models (Supplementary Data 1).

### Characterization of the insulin-resistant proteome

Proteomic analysis revealed that more than one third (37%) of proteins were significantly changed in at least one model of insulin resistance (Dunnett's post-hoc adj. $p < 0.05$, Model/CTRL > 1.5, 1398 up-regulated and 1410 down-regulated proteins, Supplementary Fig. 1i, Supplementary Data 1). This included 1693 differentially expressed proteins in CI, 1621 in DEX and 1019 in TNF (Supplementary Fig. 1j, k, Supplementary Data 1). In contrast, only 9 proteins had altered expression levels in response to MPQ and 9 in response to AA (6 common to both). The absence of widespread proteome changes in these conditions reflects the acute treatment duration (2 h), since in contrast to the phosphoproteome the proteome requires between 3–6 h to enact dynamic expression changes[24]. Proteomic changes in CI, DEX and TNF were consistent with those previously reported[3]. For example, GLUT4 (SLC2A4) and AKT2 were down-regulated (Supplementary Fig. 1k, Supplementary Data 1), and proteins in pathways such as peroxisome proliferator-activated receptor signaling, insulin signaling, lipid metabolism (fatty acid metabolism, biosynthesis of unsaturated fatty acids, the mevalonate/terpenoid backbone biosynthesis pathway, cholesterol biosynthesis), branched chain amino acid metabolism (valine, leucine, and isoleucine degradation) and oxidative phosphorylation were altered (Supplementary Fig. 2a, b). In general, these pathways also behave similarly in the transcriptome of related models[25]. To identify functionally associated proteins with altered expression in insulin resistance, we extracted protein-protein interaction (PPI) data from STRING[26] for networks comprising proteins up- or down-regulated in two or more models (388 up-regulated, 621 down-regulated). This revealed PPI clusters enriched in spliceosomal (up-regulated), endocytic (up-regulated) and ribosomal (up- and down-regulated) proteins, suggesting that these protein networks are sensitive to multiple perturbations that elicit insulin resistance (Supplementary Fig. 2c, d, Supplementary Data 2). Thus, our deep proteome analysis confirms previous reports of extensive changes in protein expression across multiple models of insulin resistance and provides a quantitative atlas for studies investigating underlying causes of insulin resistance.

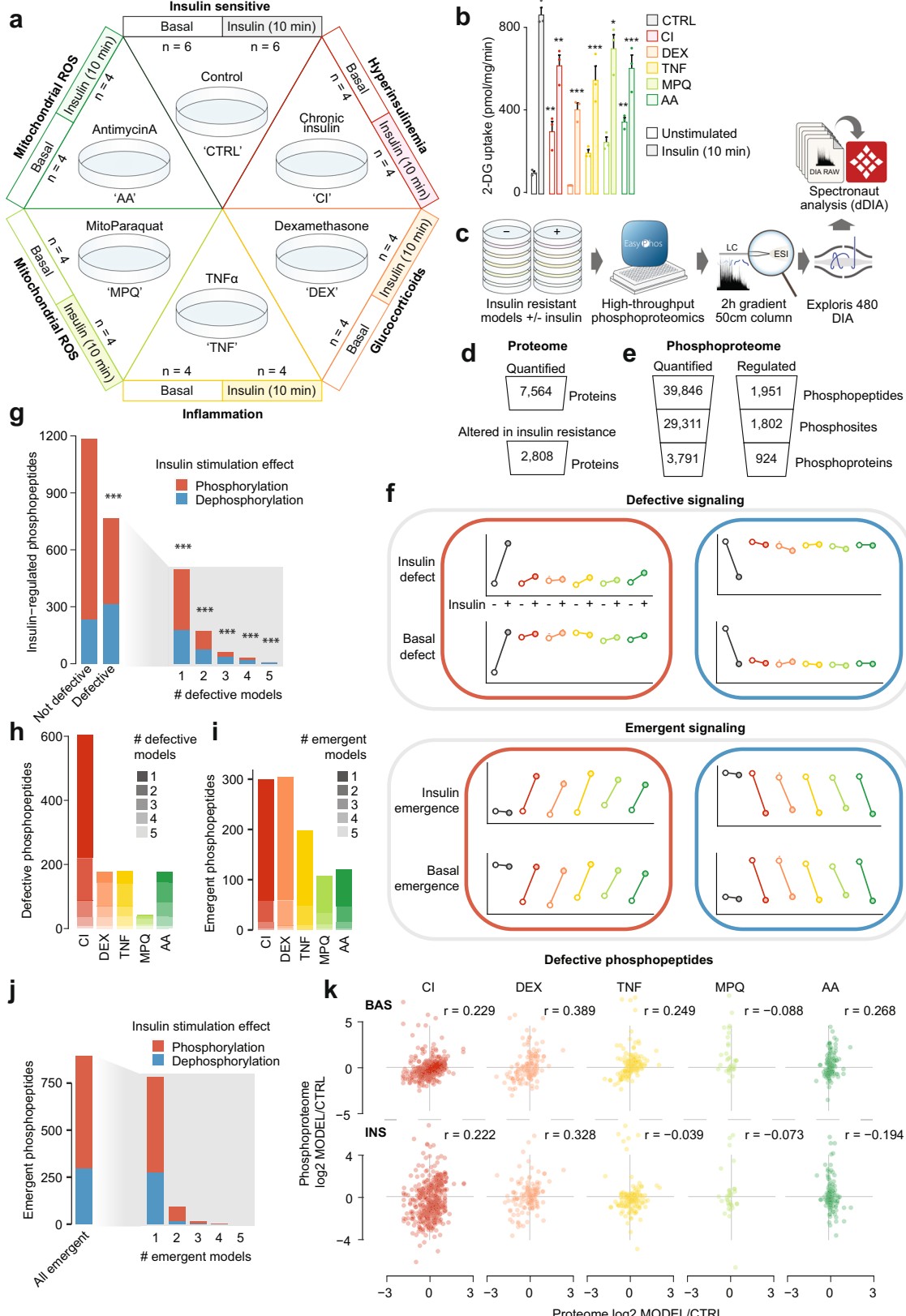

### Insulin resistance rewires the insulin signaling network

The absence of widespread changes to the proteome in our acute models (MPQ, AA; 2h) emphasizes that insulin resistance in these models is unlikely to be driven at the level of protein expression. This further prompted us to consider changes to global protein phosphorylation networks, which have been shown to operate on much

faster timescales than both transcriptional and translational machinery[24]. In our phosphoproteomics data we detected 1,951 insulin-regulated phosphopeptides in control cells that had altered abundance following insulin treatment (log2 fold-change > 0.58 or < -0.58, adj. $p < 0.05$) (Fig. 1e, Supplementary Data 3). Among these phosphopeptides were numerous known insulin-responsive

**Fig. 1 | Analysis of the proteome and phosphoproteome of insulin-resistant adipocytes. a** Adipocyte insulin resistance models and sample sizes. **b** [3]H-2DG uptake into insulin resistant adipocytes at 37 °C after 10 min Data analyzed by two-way ANOVA corrected for multiple comparisons (Dunnett's test), control vs insulin-resistant models in unstimulated or stimulated cells (*). Data are presented as mean values +/- S.E.M. n = 3 biologically independent samples. *p-values (left-to-right): 0.0092, 0.0015, < 0.0001, < 0.0001, 0.0414, 0.0014, 0.0009. **c** Workflow for 3T3-L1 adipocyte insulin resistance phosphoproteomes. Quantification from **d** proteomic and **e** phosphoproteomic analysis of insulin-resistant adipocytes. "Altered in insulin resistance" refers to proteins changed in ≥ 1 models compared to control, and "Regulated" to phosphopeptides regulated by insulin in control cells. **f** Theoretical depiction of two ways in which insulin-stimulated protein phosphorylation can change in insulin resistance. **g** Distribution of insulin resistance defects among phosphopeptides regulated by insulin in control cells. Phosphopeptides are separated by whether they are defective in at least one insulin resistance model ('Defective'). The maximum number of models in which each phosphopeptide is defective is shown. One-sided Fisher's exact tests were performed to assess whether phosphopeptides defective in at least 1, 2, 3, 4, or 5 models were more likely to be dephosphorylated in control cells (p-values corrected by Benjamini-Hochberg, *p-values (left-to-right): 1.41e[−23], 1.41e[−23], 5.89e[−17], 3.50e[−14], 2.33e[−09], 0.000136 (left-to-right)). **h** Number of insulin-regulated phosphopeptides defective in each model. **i–j** The distribution of emergent phosphopeptides. **k** Correlation of proteome and unstimulated ('BAS') or insulin-stimulated ('INS') phosphoproteome changes in phosphopeptides with defective phosphorylation in the indicated models. r = pearson's correlation coefficient. *0.01 < p < 0.05, **0.001 < p < 0.01, ***p < 0.001. Source data are provided as a Source Data file.

phosphorylation sites serving as positive controls. These included AKT1S1 T247, RPS6 S235/236, MTOR S2481, MAPK1 Y185, MAPK3 T203/Y205, GYS1 S641 and TBC1D4 S595 (Supplementary Fig. 3a, Supplementary Data 3).

We next considered how changes in protein phosphorylation could contribute to insulin resistance (Fig. 1f). One mode is 'defective phosphorylation', whereby insulin increases or decreases the abundance of a phosphosite in control cells but fails to do so in insulin resistance. As insulin rapidly modulates protein phosphorylation to enact its functional outcomes such as GLUT4 translocation, defects in insulin-regulated phosphorylation could lead to insulin resistance. In the case of a phosphosite normally increased by insulin, a defect could be due to the failure of insulin to increase the phosphosite to appropriate levels (insulin defect), or because the phosphosite is already elevated prior to the addition of insulin (basal defect) (Fig. 1f). Another mode of dysregulation we considered is 'emergent phosphorylation', whereby a phosphosite is not altered by insulin in control conditions but becomes regulated by insulin only in insulin resistance (Fig. 1f). Emergent signaling events that counteract the canonical functions of insulin could contribute to the etiology of insulin resistance.

We first examined defective phosphorylation. Remarkably, 767 of the 1,951 insulin-responsive phosphopeptides detected in control cells were not regulated by insulin in one or more insulin-resistant models (Fig. 1g). The extent of these defects varied substantially between models, from 604 phosphopeptides dysregulated in CI, to only 43 in MPQ (Fig. 1h). Only 128 of these 1951 phosphopeptides were defective in three or more models, and just seven were defective in all models (Supplementary Data 3, Fig. 1g). This supports the view that a given insult may cause many changes to signaling networks not specific to insulin resistance, highlighting the utility of studying multiple models. Although we observed widespread defects in insulin signaling, even more phosphopeptides displayed an emergent response to insulin in one or more models (896 phosphopeptides, Fig. 1i, j). This indicates that insults causing insulin resistance do not just attenuate insulin signaling, rather they rewire the insulin-responsive signaling network. Herein we define "rewiring" as the selective, potentially reversible weakening or strengthening of multiple connections in the insulin signaling network. Emergent signaling events were less conserved between models compared to defective signaling, with only 19 sites emergent in three or more models (Fig. 1i, j).

Comparative analysis of our deep proteomes and phosphoproteomes provides the opportunity to explore whether altered signaling coincided with changes in protein abundance. To this end, we correlated phosphopeptide and protein abundance changes in each model compared to control cells for defective phosphopeptides (Fig. 1k) or emergent phosphopeptides (Supplementary Fig. 3b). In all models, and regardless of insulin stimulation status, we observed poor correlation between changes in phosphopeptide and protein expression. This suggests that signaling changes were largely independent from changes in kinase-substrate kinetics driven by alterations in substrate protein expression. Overall, our findings imply that insulin resistance substantially rewires the insulin signaling network, and this is generally not due to equivalent changes in protein levels.

## Profiling kinase regulation in insulin resistance

Mechanistically, changes in signaling observed in insulin resistance could be caused by the dysregulation of one or more protein kinases. To identify potential kinases involved, we performed kinase substrate enrichment analysis (KSEA)[27], which assessed changes in in vivo effective kinase activity across the insulin resistance models using the insulin response of previously annotated kinase substrates (PhosphositePlus[28], Fig. 2a). KSEA recapitulated the known activation profiles of multiple kinases in insulin-responsive control cells, including AKT, mTOR, p70S6K, JNK, and ERK[29,30]. p70S6K is activated by mTOR in response to insulin, and the activation profiles of both kinases were highly correlated across models, indicating that our analysis accurately captured kinase regulatory patterns. However, this approach is constrained to kinases with an abundance of high-quality, context-relevant substrate annotations. This is exemplified by the fact that GSK3 was erroneously reported by KSEA to be activated by insulin in control cells, while it is known that insulin inhibits GSK3 through phosphorylation by AKT[31] at key sites. Indeed, phosphorylation of these canonical AKT inhibitory sites on GSK3α (S21) and GSK3β (S9) were robustly increased in response to insulin in control cells (Fig. 2b). As PhosphositePlus annotations are aggregated from different experimental designs (for example in vitro and in vivo assays) and from different systems (for example different cell lines and conditions), we surmised that a subset of the annotated GSK3 substrates were not regulated by GSK3 in our experimental system. This underscores the importance of mapping high-quality kinase substrates relevant to a particular biological context.

Two kinases displayed impaired insulin-activation in three or more insulin resistance models, ERK (CI, TNF, AA), and CDK5 (CI, TNF, MPQ, AA, Fig. 2a). Accordingly, insulin-stimulated phosphorylation of the MAPK3/ERK1 and MAPK1/ERK2 activation sites were significantly attenuated in CI, TNF, and AA compared to CTRL (Supplementary Fig. 4a). These results support previous findings that ERK activation is either defective[32] or intact[33] in skeletal muscle insulin resistance, but do not support the common view that insulin resistance is mediated by ERK hyperactivity[34–37]. CDK5 activation has been implicated in insulin-stimulated glucose uptake[38], supporting a putative role for CDK5 dysregulation in insulin resistance. Collectively, this analysis suggests impaired insulin-activation of ERK and CDK5 may contribute to insulin resistance.

## Characterizing defective and emergent insulin signaling

Most studies reporting altered insulin signaling during insulin resistance have focused on canonical insulin signaling intermediates such as AKT[1,16]. Our global analysis did reveal that several phosphopeptides from proteins considered part of the canonical 'insulin signaling pathway' (defined in Supplementary Data 4) were defective across multiple models of insulin resistance, including S502 on TSC1 and S244

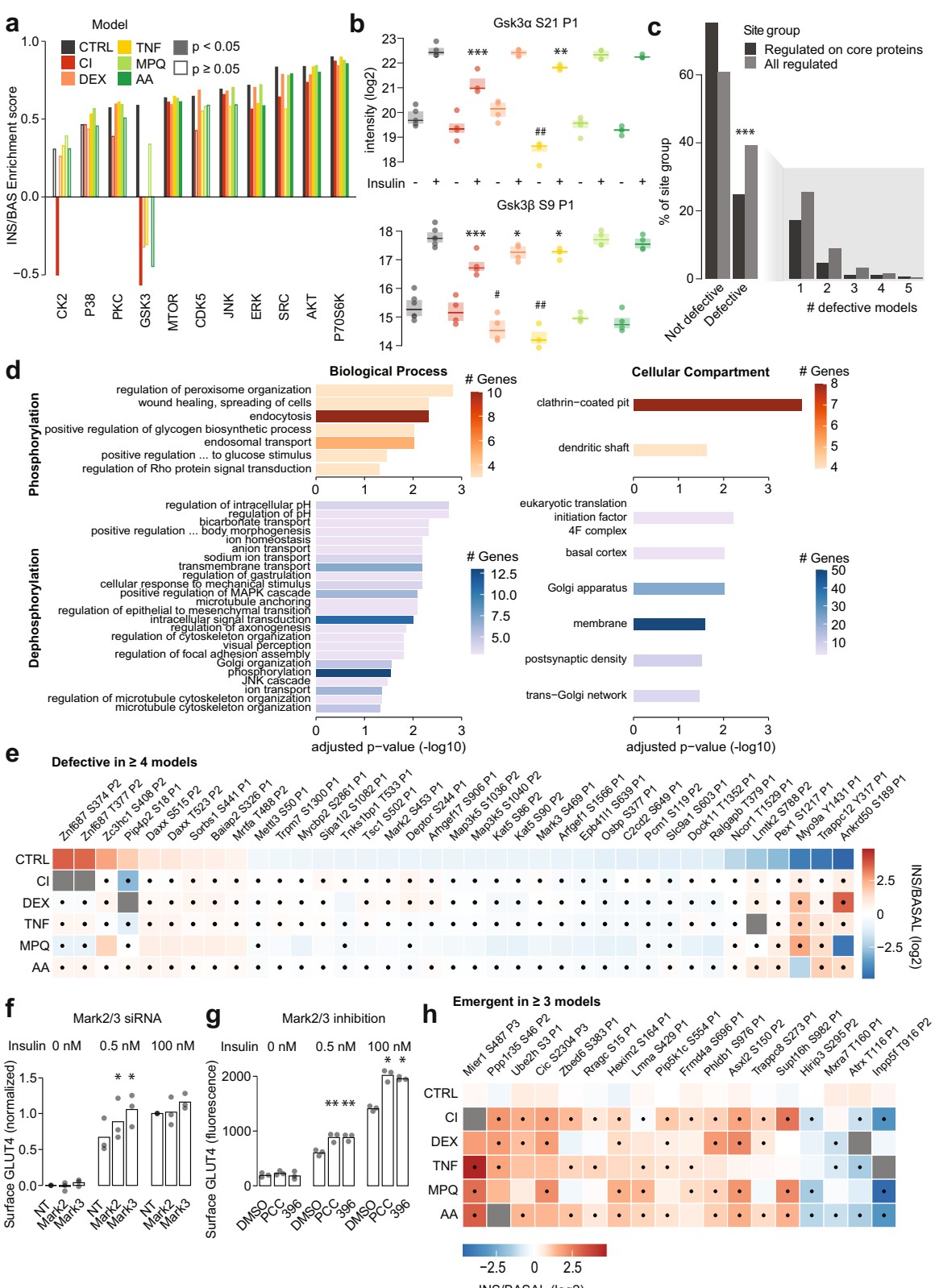

on DEPTOR (Supplementary Fig. 4b). However, KSEA revealed that AKT, the key regulator of insulin-stimulated GLUT4 translocation and glucose transport[29], and downstream kinases mTOR and p70S6K, were activated to a similar degree in control and insulin-resistant cells (Fig. 2a). Moreover, insulin signaling to proteins comprising the canonical insulin signaling pathway remained intact in insulin-resistant cells, as insulin-regulated phosphopeptides from these proteins were

1.7 times less likely to be defective than those on proteins outside of this pathway (Fig. 2c, Supplementary Fig. 4c, one-sided Fisher's exact test: $p = 6.73e^{-9}$). Overall, dysregulation within the core insulin signaling pathway was limited.

We next considered broader characteristics of the insulin resistance signaling network. Phosphopeptides defective in two or more models (271 phosphopeptides; 211 proteins) were enriched for

**Fig. 2 | Impaired and emergent signaling in insulin resistance. a** KSEA[27] was performed in insulin resistance models using log2 INS/BAS fold changes. Only kinases regulated ($p < 0.05$) in ≥1 model are shown. **b** Intensity of GSK3α S21 and GSK3β S9 phosphopeptides. "P1" denotes that phosphopeptides contained only one localized phosphosite. ANOVAs and two-sided Dunnett's post hoc tests were performed on unstimulated (#) or insulin-stimulated (*) phosphoproteome data, comparing insulin resistant models to control cells. $n = 4–6$ independent biological replicates. $P$-values (left-to-right, top-to-bottom): #0.000222, 0.0407, 0.00725. *$2.623^{-6}$, 0.00269, 0.000236, 0.0337, 0.0316 (**c**) Distribution of phosphorylation defects among regulated phosphopeptides on all proteins or on proteins considered part of the core insulin signaling pathway. Regulated phosphopeptides on core insulin signaling protein are less likely to be defective than expected (one-sided Fisher's exact test, *$p = 6.74e^{-9}$). **d** Enriched pathways (GO-term, one-sided Fisher's exact test, Benjamini-Hochberg adjustment) containing insulin up-regulated (red) or down-regulated (blue) phosphopeptides defective in ≥2 models. Colour indicates the number of phospho-dysregulated genes. **e** Phosphopeptides

that were defective in ≥4 insulin resistance models. Dots indicate that model insulin responses were significantly different to control insulin responses (adj. $p < 0.05$). **f** 3T3-L1 adipocytes were transfected with non-targeting siRNA ('NT') or siRNA targeting *Mark2* or *Mark3*. After 20 min insulin stimulation, plasma membrane GLUT4 was measured and normalized (see Methods). *Mark2* and *Mark3* siRNA were compared to NT at 0.5 nM and 100 nM insulin by two-way ANOVA and Holm-Šídák post-hoc tests (*). *$p$-values (left-to-right): 0.0481, 0.0481. **g** 3T3-L1 adipocytes pretreated for 90 min with DMSO, 0.5 μM PCC020817 (PCC) or 10 μM 39621 (396) were stimulated with insulin for 20 min, following plasma membrane GLUT4 measurement. PCC and 396 were compared to DMSO at 0.5 nM and 100 nM insulin by two-way ANOVA and Holm-Šídák post-hoc tests (*). *$p$-values (left-to-right): 0.0055, 0.0038, 0.0150, 0.0119. **h** Phosphopeptides emergent in ≥3 models. Ppp2r5b S13 was excluded to optimize the heatmap scale and is shown in Supplementary Fig. 4h. */#$0.01 < p < 0.05$, **/##$0.001 < p < 0.01$, ***/###$p < 0.001$. Source data are provided as a Source Data file.

proteins in processes such as "endocytosis", "focal adhesion assembly", and "transmembrane transport", and in cellular compartments such as the "golgi apparatus" and "membranes", highlighting these as potential sites of insulin resistance (Fig. 2d). While control cells exhibited a greater proportion of increased protein phosphorylation than decreased phosphorylation (1402 increased phosphopeptides, 459 decreased phosphopeptides), insulin-regulated dephosphorylation was preferentially impaired in insulin resistance (Fig. 1g). Specifically, protein dephosphorylation was 1.8 times more likely to be defective in one or more models compared to phosphorylation (one-sided Fisher's exact test: adj. $p = 1.41e^{-23}$), and 4.6 times more likely to be defective in three or more models ($p = 3.50e^{-14}$, Fig. 1g). These instances of defective dephosphorylation were generally caused by changes in the insulin-stimulated phosphoproteome (i.e., an insulin defect) rather than the unstimulated phosphoproteome (a basal defect) (Fig. 1f, Supplementary Fig. 4d). This was also observed for defective phosphorylation (Fig. 1f, Supplementary Fig. 4d). Together, these data suggest that impaired decreases in phosphorylation is a defining feature of insulin resistance.

Finally, we rationalized that proteins with phosphosites defective across most models would be likely to contribute to insulin action and resistance. The 37 phosphopeptides defective in four or more models included S453 and S469 on the kinases MARK2/3 (Fig. 2e). These phosphosites correlated positively with a subset of MARK2/3 substrates (Supplementary Fig. 4e) and have high phosphosite functionality scores as established by Ochoa et al.[39] (0.68 and 0.58) indicating they may enhance MARK2/3 activity. As these sites were decreased by insulin in CTRL cells, we hypothesized that MARK2/3 may negatively regulate GLUT4 translocation. Consistent with this, siRNA-mediated knockdown of *Mark2/3* (approximately 50% reduction, Supplementary Fig. 4f) or pharmacological inhibition of MARK2/3 (confirmed by reduced phosphorylation of the substrate HDAC, Supplementary Fig. 4g) increased insulin-stimulated GLUT4 translocation (Fig. 2f, g). These data agree with the insulin hypersensitivity observed in the adipose tissue of MARK2 knock-out mice[40], and highlight S453/S469 as potential regulatory nodes through which MARK2/3 may contribute to insulin action.

Several other phosphopeptides defective in four or more models had known regulatory roles including S86 on Kat5, which promotes autophagy[41] and apoptosis[42], and S1040 on the stress and reactive oxygen species-sensitive kinase Map3k5/Ask1, which inhibits Ask1 activity and apoptosis[43] (Fig. 2e). Genetic variants in Ask1 are associated with insulin resistance and type 2 diabetes in Pima Indians[44], further supporting a role for Ask1 in insulin resistance. We also observed sites of unknown function on proteins potentially involved in GLUT4 trafficking, including S411 on Sorbs1, a signaling adaptor that links the activated insulin receptor to lipid rafts and promotes GLUT4 translocation[45]; S603 on Slc9a1, a Na H+ exchanger that promotes actin

rearrangement following activation by insulin[46]; T533 on Tnks1bp1 which is an interactor of Tankyrase, an insulin-activated GLUT4 vesicle-interactor required for GLUT4 translocation[47,48]; and Y317 on Trappc12 which is part of TRAPP, a complex involved in ER to Golgi trafficking[49]. There were 19 emergent sites in three or more models, including an uncharacterised site on pleckstrin homology-like domain family B member 1 (PHLDB1), a regulator of insulin-stimulated GLUT4 translocation[50] (Fig. 2h, Supplementary Fig. 4h). Proteins possessing sites with emergent phosphorylation in three or more models were enriched in the GO terms "negative regulation of transcription and RNA polymerase II"; and "negative regulation of transcription, DNA-templated", suggesting altered transcriptional regulation is a key target of signaling changes in insulin-resistant cells (Supplementary Fig. 4i).

In all, rewiring of the insulin signaling network was largely exclusive of well-studied insulin signaling proteins. Instead, changes observed comprised a set of kinases, proteins, and pathways without known links with insulin at present. This suggests our knowledge of the extent of mechanistic insulin signaling remains far from complete. Our data also imply that multiple parallel signaling alterations may cumulatively contribute to insulin resistance. If this is the case, targeting a single kinase or pathway may only confer a partial benefit to insulin sensitivity, pointing towards the benefits of a polypharmacological approach in drug discovery for insulin resistance.

## Targeted analysis of the de-phosphorylation defect in adipocytes

We observed that insulin-regulated dephosphorylation was widely dysregulated in all 3T3-L1 insulin resistance models (Fig. 1g). Acute phosphatase inhibition impaired insulin-stimulated 2DG uptake in 3T3-L1 adipocytes and adipose explants, and HA-GLUT4 translocation in 3T3-L1 adipocytes (Supplementary Fig. 4j–l) as previously reported[51], supporting the notion that protein dephosphorylation is important to insulin action. However, we detected no change in global phosphatase activity in the CI, DEX, or TNF models (Supplementary Fig. 4m), and changes in the expression of specific phosphatases were limited to chronic models of insulin resistance (CI, DEX, TNF) and not acute models (AA, MPQ) (Supplementary Fig. 4n). Therefore, we hypothesized that the defective insulin-stimulated protein dephosphorylation observed in multiple models of insulin resistance may be driven by aberrant kinase deactivation by insulin.

One of the best-studied kinases inactivated by insulin is glycogen synthase kinase 3 (GSK3). Under normal conditions, GSK3 is constitutively active, and its phosphorylation by AKT at S9/21 inactivates the kinase[31]. Several bona fide GSK3 substrates displayed impaired dephosphorylation across insulin resistance models, such as S641 on glycogen synthase (CI and AA), and S86 and S90 on the histone acetyltransferase Kat5 (CI, DEX, TNF, and AA) (Supplementary Fig. 5a). In

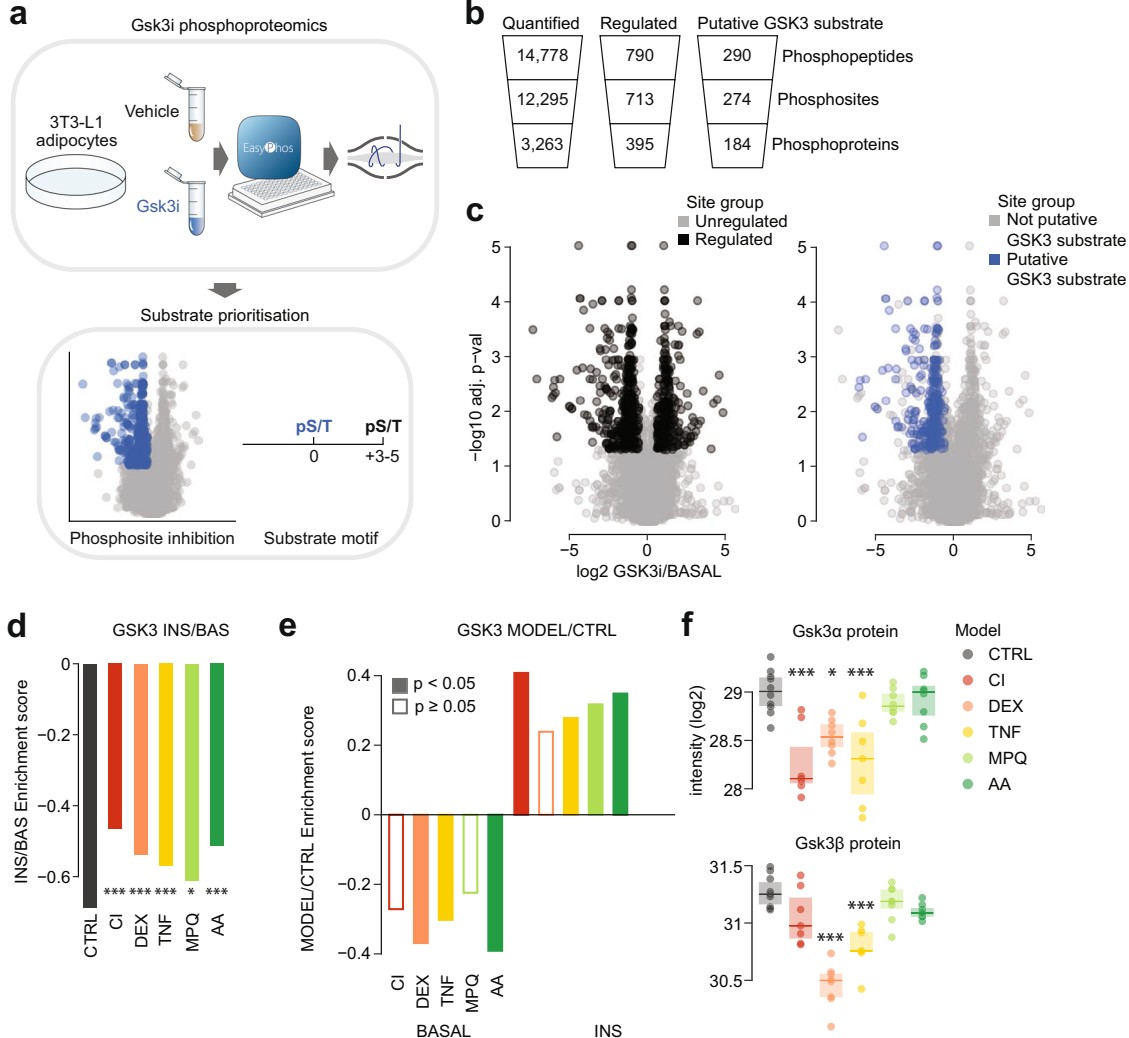

**Fig. 3 | Insulin-regulated inhibition of the kinase GSK3 is impaired in insulin resistance. a** Pipeline for identification of context-specific kinase substrates, exemplified by GSK3 in 3T3-L1 adipocytes. **b** Quantification from GSK3 inhibitor phosphoproteomics. **c** The effect of 20 min GSK3 inhibition on the 3T3-L1 adipocyte phosphoproteome was tested by two-sided t-tests followed by Benjamini-Hochberg p-value adjustment. Black indicates phosphopeptides significantly up-regulated or down-regulated by GSK3 inhibition by more than 1.5-fold and blue indicates down-regulated phosphopeptides matching the GSK3 substrate motif. **d** KSEA in each insulin resistance model using log2 INS/BAS fold changes and putative GSK3 substrates decreased in CTRL cells (log2 INS/BAS < 0). To examine differences between control cells and models, KSEA[27] was performed using log2

INS/BAS fold changes normalized to control cells (*). *p*-values (left-to-right): 0, 0, 0, 0.049, 0. **e** KSEA[27] in unstimulated ('BASAL') and insulin-stimulated ('INS') phosphoproteomes using log2 Model/CTRL fold changes and the same substrates as in **d**. Filled-in columns indicate significant changes in GSK3 activity (*p* < 0.05). P-values (left-to-right): 0.165, 0.001, 0.006, 0.542, 0, 0, 0.069, 0.017, 0.004, 0. **f** Intensity of GSK3α/β total protein with significant differences compared to control cells indicated (ANOVA, two-sided Dunnett's post hoc tests, Benjamini-Hochberg adjustment, *). *p*-values (left-to-right, top-to-bottom): 0.0000475, 0.0105, 0.000154, 2.32e$^{-11}$, 1.98e$^{-5}$. *0.01 < *p* < 0.05, **0.001 < *p* < 0.01, ***p* < 0.001. *n* = 4–6 independent biological replicates. Source data are provided as a Source Data file.

contrast, while PKA is also inactivated by insulin, we found only one annotated PKA substrate with defective dephosphorylation (Sik3 S493), which was only defective in a single model (DEX, Supplementary Fig. 5b). These observations suggested that dysregulation of GSK3 activity may contribute to impaired dephosphorylation in insulin resistance.

As noted, our earlier KSEA using existing database annotations failed to accurately assess GSK3 activity (Fig. 2a). Therefore, we took this opportunity to establish a pipeline to identify a high-quality list of putative GSK3 substrates in adipocytes (Fig. 3a), reasoning that this knowledge would facilitate a more accurate evaluation of GSK3 activity in insulin resistance. To this end, we generated a phosphoproteome in 3T3-L1 adipocytes comprising 14,778 phosphopeptides. Of these, 790 (on 395 proteins) were responsive to acute treatment with the highly selective GSK3 inhibitor LY2090314 (hereafter referred to as "GSK3i")[52], with 309 phosphopeptides increasing and 481 decreasing

by at least 1.5-fold (adj. *p* < 0.05, log2 fold-change > 0.58 or < −0.58, respectively) (Fig. 3b, c). To enrich these data for sites more likely to be direct substrates of GSK3 in adipocytes (as opposed to downstream of GSK3), we selected only those that were down-regulated in response to GSK3i and also contained the motif of a GSK3 substrate (pS/T X(2-4) pS/T, where the first S/T is phosphorylated by GSK3 and the second is a priming phosphosite). The resulting 290 phosphopeptides corresponded to 274 phosphosites on 184 proteins (Fig. 3b, c, Supplementary Data 5), and contained previously identified GSK3 targets such as S641, S645, and S649 on glycogen synthase[53], and S514 and S518 on DPYSL2[54,55], confirming the validity of our approach.

## Insulin-regulated inhibition of the kinase GSK3 is impaired in insulin resistance

To utilize this atlas of adipocyte-specific GSK3 substrates to analyze our insulin signaling data, we next mapped the 274 putative

GSK3 substrate phosphosites onto our 3T3-L1 insulin resistance phosphoproteome. We found a high overlap between these datasets, identifying 191 corresponding phosphosites that decreased in abundance in response to insulin ($\log_2$ insulin/basal < 0), further supporting that these sites were regulated by GSK3 in adipocytes. Performing a KSEA now using these sites revealed that GSK3 was inactivated in response to insulin in control cells in our original phosphoproteomics study as expected, and remarkably, that this inactivation was attenuated across all insulin resistance models (Fig. 3d, Supplementary Fig. 5c). Putative GSK3 substrates with defective dephosphorylation in insulin resistance included S520 and S553 on PHLDB1; S350, S354, and S358 on the microtubule regulator SLAIN2; and S1036 on ASK1 (Supplementary Fig. 5d). Of note, the ASK1 inhibitory site S1040 is the priming site for S1036, and both sites display defective dephosphorylation in four 3T3-L1 models (Fig. 2e). These sites, therefore, represent an unappreciated intersection between GSK3 and ASK1 signaling that is altered in insulin resistance. In addition, the GLUT4 translocation-regulator PHLDB1 contained a site that was emergent in 4 models (Fig. 2h), marking PHLDB1 as a node of crosstalk for signaling pathways both attenuated and promoted in insulin resistance. Our context-specific kinase substrate profiling suggests that the deactivation of GSK3 by insulin is impaired in multiple models of insulin resistance, and that this may contribute to the impaired protein dephosphorylation, and defective insulin-stimulated glucose transport, observed across these models.

Impaired insulin-stimulated GSK3 deactivation could be driven by lower unstimulated activity or elevated insulin-stimulated activity, or both. To distinguish these possibilities, we performed KSEA on phosphopeptide abundance relative to control cells, under both unstimulated and insulin-stimulated conditions. In unstimulated cells, DEX, TNF and AA had reduced GSK3 activity relative to control, while in insulin-stimulated cells, CI, TNF, MPQ and AA had increased GSK3 activity relative to control (Fig. 3e). Overall, more models had altered GSK3 activity in the insulin-stimulated state, suggesting that the primary defect of GSK3 regulation in insulin resistance is the inability of insulin to attenuate GSK3 activity.

We next asked whether changes in the abundance of GSK3α and GSK3β isoforms and/or regulatory phosphorylation of GSK3 could explain the observed changes in GSK3 activity. GSK3 protein expression was decreased in most chronic 3T3-L1 models (Fig. 3f, Supplementary Fig. 5e; GSK3α: CI, DEX, and TNF; GSK3β: DEX, TNF). However, the same changes were not observed in acute models (MPQ, AA), despite impaired insulin-dependent GSK3 signaling (Fig. 3f). In some cases, impaired attenuation of GSK3 activity by insulin may be partially driven by impaired phosphorylation of the AKT inhibitory sites, as insulin-mediated phosphorylation of GSK3α S21 was blunted in CI and TNF models, as was insulin-mediated GSK3β S9 phosphorylation in CI, DEX, and TNF (Fig. 2b, Supplementary Fig. 5e). However, normalizing phosphorylated GSK3 to the total protein abundance suggested that these changes in S21 or S9 phosphorylation in DEX and TNF could potentially be explained by decreased expression of GSK3, while the changes in CI were additionally mediated by decreased AKT signaling to GSK3 (Supplementary Fig. 5f). Notably, as inhibitory phosphorylation and protein abundance were decreased to a similar extent in TNF, inhibitory phosphorylation should not explain why insulin-stimulated GSK3 activity was increased in TNF relative to control. Phosphorylation of GSK3α/β at Y279/Y216 has been reported to enhance kinase activity and may be an autophosphorylation event[56]. We identified no significant insulin response in GSK3α Y279, and GSK3β Y216 was not quantified (Supplementary Fig. 5g). However, GSK3α Y279 was weakly decreased by insulin in CTRL cells but not decreased in DEX, TNF, and AA. Although we cannot exclude the involvement of tyrosine phosphorylation in impaired deactivation of GSK3, the small magnitude of Y279 changes between models suggests that any

contribution to dysregulated GSK3 function may be minor. Although these alterations in protein expression and signaling to GSK3 may, to some extent, explain the behavior of GSK3 in our chronic models (CI, DEX, and TNF), they do not sufficiently explain the impairment in GSK3 inactivation observed in our acute models (MPQ, AA). This suggests that aside from protein abundance and insulin signaling to GSK3, other factors that control GSK3 are likely disrupted during insulin resistance.

## Signaling rewiring and GSK3 dysregulation in insulin-resistant adipose tissue

We next assessed if key findings from 3T3-L1 adipocytes, including impaired GSK3 regulation, were recapitulated in in vivo models of insulin resistance. To this end, we fed mice either a chow diet (**CHOW**) or a high-fat high sucrose diet (HFD) for 14 d (insulin resistant (**HFD**))[3]. For a third group of mice fed HFD for 14 d, we switched the diet back to chow for a further 5 d (reversal of insulin resistance (**REV**), Fig. 4a). In the mice fed a 14 d HFD, insulin-stimulated 2DG into epididymal adipose tissue was reduced compared to the CHOW group, while returning mice to a chow diet for 5 days improved insulin sensitivity by approximately 40% (Fig. 4b).

To assess insulin signaling we performed phosphoproteomic analysis of epididymal adipose tissue from CHOW, HFD and REV mice treated with saline or insulin (Fig. 4c, Supplementary Fig. 6a–d). We identified 319 insulin-regulated phosphopeptides on 210 proteins in CHOW mice (Fig. 4c, Supplementary Data 6). Insulin signaling was substantially rewired in HFD mice, as 203 of these 319 phosphopeptides were no longer insulin-responsive in HFD, while a separate set of 105 phosphopeptides displayed emergent insulin-regulation in HFD (Fig. 4d, Supplementary Fig. 6e, Supplementary Data 6). Akin to insulin-resistant 3T3-L1 adipocytes, the majority of the 203 defective phosphopeptides were outside core insulin signaling pathways (Fig. 4e), and emergent phosphorylation was enriched in transcriptional regulators (Supplementary Fig. 6f). We next assessed whether GSK3 activity was altered in insulin-resistant adipose tissue, using our experimentally-defined list of potential GSK3 substrates. KSEA on the 168 GSK3 substrates downregulated by insulin in CHOW tissue (CHOW $\log_2$ INS/BASAL < 0) revealed that insulin-stimulated GSK3 deactivation was impaired in HFD and partially restored in REV tissue (Fig. 4f). As adipose tissue insulin sensitivity was also impaired in HFD and partially restored in REV (Fig. 4b), this supports the notion that insulin regulation of GSK3 is important for insulin sensitivity. GSK3 substrates with defective dephosphorylation in HFD tissue included S72 on TRARG1 (Supplementary Fig. 6g). Trarg1 has been implicated in GLUT4 trafficking[57,58], and the dephosphorylation of this site may promote GLUT4 translocation[59], so Trarg1 may link GSK3 dysregulation to impaired GLUT4 trafficking in insulin resistance.

As in 3T3-L1 adipocyte models of insulin resistance, the dysregulation of GSK3 in insulin-resistant tissue was due to a combination of altered activity before and after insulin stimulation (Fig. 4g). In particular, before insulin stimulation, GSK3 activity was decreased in HFD and REV tissue compared to CHOW (Fig. 4g), which may be due to increased inhibitory phosphorylation of S9 on GSK3β (Fig. 4h). Phosphorylation of S21 on GSK3α was not adequately quantified, and phosphorylation of Y279 on GSK3α was not changed by insulin (Supplementary Fig. 6h). After insulin stimulation GSK3 activity was elevated in HFD (Fig. 4g), however this cannot be explained by S9 phosphorylation as this was equivalent to that observed in adipose tissue from CHOW-fed mice (Fig. 4h). In addition, these changes in GSK3 activity were more pronounced in insulin-stimulated tissue compared to unstimulated tissue, supporting the observation we made in 3T3-L1 adipocytes that GSK3 dysregulation is mainly due to the inability of insulin to lower its activity. Collectively, phosphoproteomic analysis of in vivo models of insulin resistance recapitulated key findings from 3T3-L1

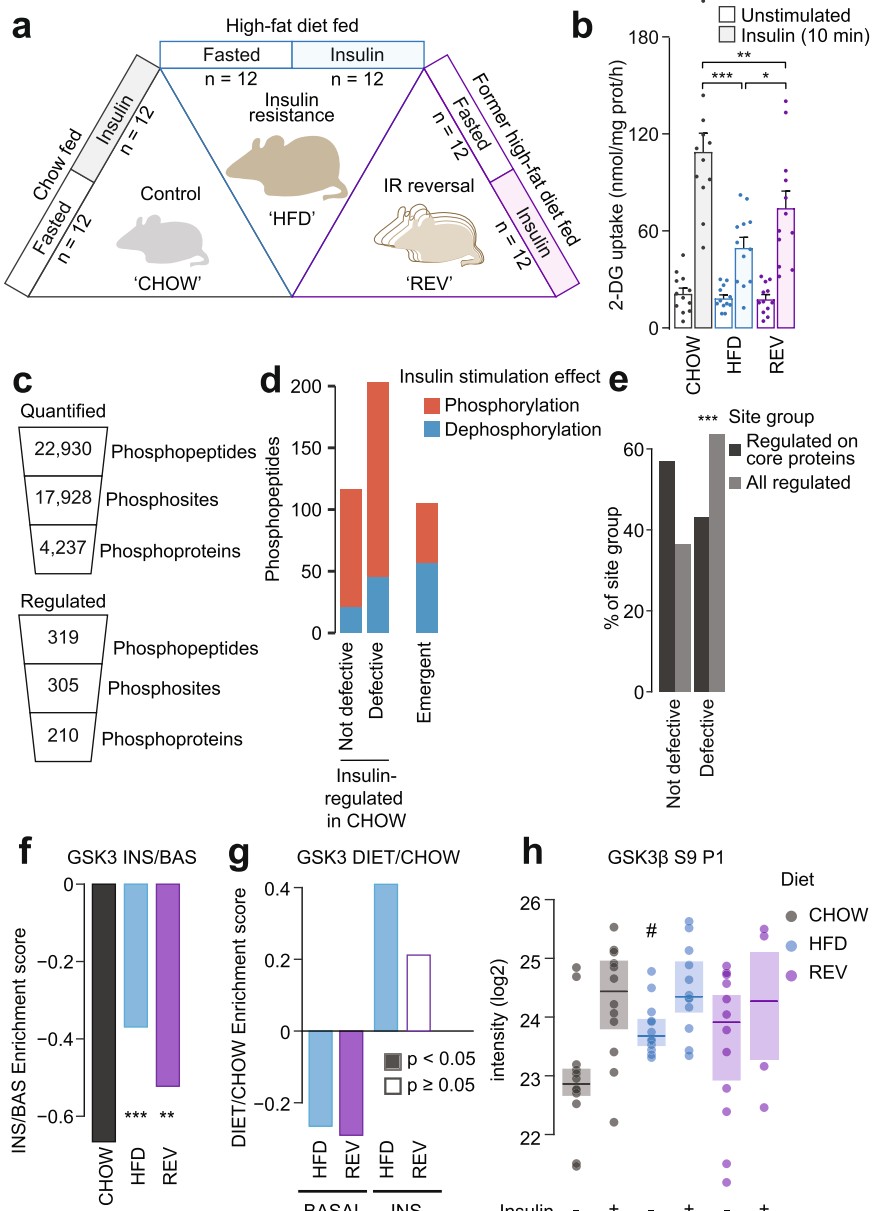

**Fig. 4 | Impaired inhibition of GSK3 in insulin-resistant adipose tissue. a** Diet regimes and sample sizes for phosphoproteomic analysis of adipose tissue. **b** [3]H-2DG uptake into epididymal adipose tissue during 10 min intraperitoneal injection with insulin or saline. Data were analyzed by two-way ANOVA corrected for multiple comparisons (Tukey's test) to compare insulin-stimulated uptake between 'CHOW', 'HFD', and 'REV' groups (*). Error bars are S.E.M. $n = 12$ biologically independent animals. *$p$-values (left-to-right): < 0.0001, 0.0026, 0.0421. **c** Quantification from phosphoproteomic analysis of adipose tissue. **d** The number of phosphopeptides that were significantly regulated by insulin in CHOW mice, further divided into sites that were defective in HFD mice. The number of phosphopeptides emergent in HFD is also shown. **e** The distribution of phosphorylation defects among regulated phosphopeptides on all proteins or proteins considered part of the core insulin signaling pathway. A one-sided Fisher's exact test was performed to assess whether regulated phosphopeptides on core insulin signaling protein were less likely to be defective than expected by chance (*$p = 4.26e^{-5}$). **f** KSEA was performed using log2 INS/BAS fold changes and putative GSK3 substrates that were decreased in CHOW mice (log2 INS/BAS < 0). KSEA[27] was performed using log2 INS/BAS fold changes in HFD and REV normalized to CHOW to assess differences between diets (*). *$p$-values (left-to-right): 0, 0.004. **g** KSEA[27] was performed in the unstimulated ('BASAL') and insulin-stimulated ('INS') phosphoproteomes using log2 DIET/CHOW fold changes and the same substrates as in **f**. Filled-in columns indicate significant changes in GSK3 activity ($p < 0.05$). $P$-values (left-to-right): 0.044, 0.008, 0, 0.265. **h** Intensity of GSK3β S9. Two-sided t-tests were performed to compare HFD or REV to CHOW in unstimulated (#$p = 0.0174$) or insulin-stimulated mice (not significant). */#$0.01 < p < 0.05$, **/##$0.001 < p < 0.01$, ***/###$p < 0.001$. Source data are provided as a Source Data file.

adipocytes, including the observation that regulation of GSK3 by insulin is impaired in insulin resistance.

## Pharmacological inhibition of GSK3 rescues insulin sensitivity in vitro and ex vivo

In our analysis of 3T3-L1 models of adipocyte insulin resistance, we detected multiple insulin signaling alterations shared across different models, leading us to hypothesize that pharmacological targeting of

any one of these alterations may only confer a partial reversal of insulin resistance. As impaired deactivation of GSK3 by insulin was observed in all insulin-resistant 3T3-L1 models and insulin-resistant adipose tissue, we decided to test this hypothesis on GSK3. We treated insulin-resistant 3T3-L1 adipocytes (CI, DEX, and TNF) with the GSK3 inhibitors GSK3i, CHIR99021 (CHIR), or AZD2858 (AZD) 1.5 h prior to administration of insulin, and monitored plasma membrane GLUT4 using an antibody that recognizes an exofacial region of GLUT4[60], or by stably

expressing an HA-GLUT4-mRuby3 reporter construct. GSK3 inhibition was confirmed by reduced phosphorylation of glycogen synthase (Supplementary Fig. 7a). All three insulin resistance treatments impaired insulin-stimulated GLUT4 translocation in response to 0.5 or 100 nM insulin in wild-type or HA-GLUT4-mRuby3-expressing cells (Fig. 5a, c, Supplementary Fig. 7b–e). GSK3 inhibition increased cell surface endogenous GLUT4 and HA-GLUT4-mRuby3 in control, TNF and DEX-treated cells. The response to GSK3 inhibition was much greater in TNF and DEX models than in control cells (Fig. 5b, d) so that in general insulin responses were nearly equivalent to control cells. However, CI-treated cells were largely refractory to GSK3 inhibition (Fig. 5a–d, Supplementary Fig. 7b–e). Notably, the effects of these GSK3 inhibitors in the DEX and TNF models were only observed in insulin-stimulated cells, suggesting that this intervention acts by specifically augmenting the insulin response.

We also examined the effect of GSK3i in explants derived from insulin-sensitive (CHOW) and insulin-resistant (HFD) mouse adipose tissue. Insulin-stimulated 2DG uptake was reduced in control DMSO-treated explants from HFD-fed mice at both 0.5 and 10 nM insulin, confirming that they were insulin resistant (Fig. 5e). Crucially, 1 h pre-treatment with GSK3i increased insulin-stimulated 2DG uptake at a submaximal insulin concentration (0.5 nM) in explants from HFD-fed mice (Fig. 5e). This effect was not observed in these explants at 10 nM insulin, nor in explants from chow-fed mice at either insulin concentration.

Previous studies have found acute and chronic inhibition of GSK3 in rodent models of insulin resistance improved insulin-stimulated glucose uptake into muscle[61–64]. Our results complement this body of research, indicating that acute inhibition of GSK3 can ameliorate adipocyte insulin resistance in diverse contexts. The reversal of insulin resistance provided by GSK3 inhibition was only partial, lending evidence to our hypothesis that insulin resistance is driven by multiple, cumulative signaling network abnormalities.

## Discussion

Here we employed phosphoproteomics to interrogate global signaling in insulin-resistant adipocytes. This is relevant to metabolic disease in humans because adipose insulin resistance is an early event in the path to type 2 diabetes, and disrupting insulin-stimulated glucose uptake in adipocytes compromises whole-body glucose homeostasis[4,5]. Previous targeted studies have associated insulin resistance with defects in proximal insulin signaling proteins, including AKT, IRS1/2, and the insulin receptor. However, the relevance of these findings has been challenged[1,16], highlighting a need for global, untargeted analyses of signaling in insulin resistance. Here we systematically address this for the first time in adipocytes. By employing standardized and controlled cell culture models spanning a broad range of known contributors to insulin resistance, we observed that the signaling underpinning phenotypic insulin resistance does not comprise a simple defect, but instead a profound rewiring of many nodes within the insulin signaling network. Pharmacologically targeting one of these nodes in isolation improved insulin sensitivity, but only partially. This highlights the utility of our approach in identifying targets to ameliorate metabolic dysfunction and supports the view that insulin resistance is a complex multi-nodal defect. Rectifying this defect will therefore likely require either the identification of common regulators of discrete defective nodes, or treatments designed to target multiple nodes ab initio (i.e., a polypharmacological approach). We verified these observations in adipose tissue from mice fed different diets, supporting the physiological relevance of our findings.

Despite substantial rewiring of insulin signaling across insulin resistance models, the regulation of canonical insulin-responsive kinases and phosphoproteins was overwhelmingly unchanged. To gain a deeper understanding of insulin resistance. It is therefore necessary to venture into the vast, non-canonical regions of insulin signaling.

However, as is the case for the majority of the phosphoproteome, most dysregulated phosphosites lack an annotated upstream kinase and have not yet been functionally characterized[65], making their role in insulin resistance difficult to disentangle. We applied two approaches to address the challenges of kinase-substrate annotation and downstream functional characterization. First, we employed a pipeline for identifying contextually-relevant substrates that uncovered hundreds of potential GSK3 substrates in adipocytes. These data subsequently facilitated the detection of GSK3 dysregulation in insulin resistance where existing database-derived annotations could not. Second, to prioritize phosphoproteins with a functional role in insulin action we a) identified phosphoproteins with dysregulated signaling across multiple models of insulin resistance; and b) combined these conserved signatures of phosphosite dysregulation with a recent computational predictor of phosphosite functionality[39]. This approach uncovered the microtubule-regulating kinases MARK2 and MARK3 as novel modulators of insulin-stimulated GLUT4 translocation. Future work should establish which substrates of MARK2/3 mediate this role, and whether they operate through the regulation of microtubule dynamics—an established step in GLUT4 trafficking[66]—or through other functions. Sustained efforts to map signaling topology and function will reveal how signaling alterations we identified in insulin resistance are embedded within and shape the insulin signaling network.

Kinases are popular targets for drug discovery efforts[67], and, as we have presented for GSK3, mapping upstream regulators of dysregulated phosphosites will identify kinases that could be pharmacologically targeted to ameliorate insulin resistance. Interestingly, there is also evidence that GSK3 hyperactivation drives dysfunction in diabetic islets[68] and that GSK3 hyperactivation in the hypothalamus impairs glucose homeostasis[69]. Thus, targeting GSK3 could provide a route to combating multi-organ dysfunction in the progression of type 2 diabetes. In this study, we acutely administered GSK3 inhibitors to restore the acute inhibition of GSK3 normally achieved by insulin, which partially restored insulin sensitivity. However, studies in GSK3α/β S21/S9A knock-in mice—which are refractory to insulin inhibition to begin with—report normal muscle insulin-stimulated glucose transport[70]. This disparity between acute pharmacological inhibition and chronic genetic activation of GSK3 may be explained by the different tissues studied (adipose vs skeletal muscle), or compensatory mechanisms invoked by the constitutive knock-in of the S21/S9A mutation. Knowledge of how GSK3 is dysregulated in insulin resistance may resolve this disparity. Protein abundance and inhibitory phosphorylation could only partially explain the changes we observed in GSK3 activity in insulin resistance, suggesting other regulatory mechanisms may be involved. Reactive oxygen species (ROS) are elevated in all models we studied[3,22,23] and ROS can crosstalk with phosphorylation through protein redox modification[71]. Moreover, GSK3β is oxidized at the functionally uncharacterized residues Cys107 and Cys76 in adipocytes following oxidative stress[71], underscoring redox modification as a potential regulator of GSK3. In contrast to GSK3 inhibition, quenching of ROS more completely reversed insulin resistance in our models[22], suggesting that ROS may dysregulate insulin signaling at several distinct nodes. To clarify and expand on these possibilities, future work should globally dissect the crosstalk between protein redox modification and phosphorylation in insulin resistance.

Here we provide a global view of the disrupted signaling that occurs during adipocyte insulin resistance and augment these data with an approach to chart dysregulated kinase signaling in a context-specific manner. Our data suggest insulin resistance may arise from the cumulative contributions of diverse dysfunctional signaling. From these we have studied GSK3 and MARK2/3, exemplifying the utility of this resource, however, these data comprise many more putative mediators that warrant future investigation.

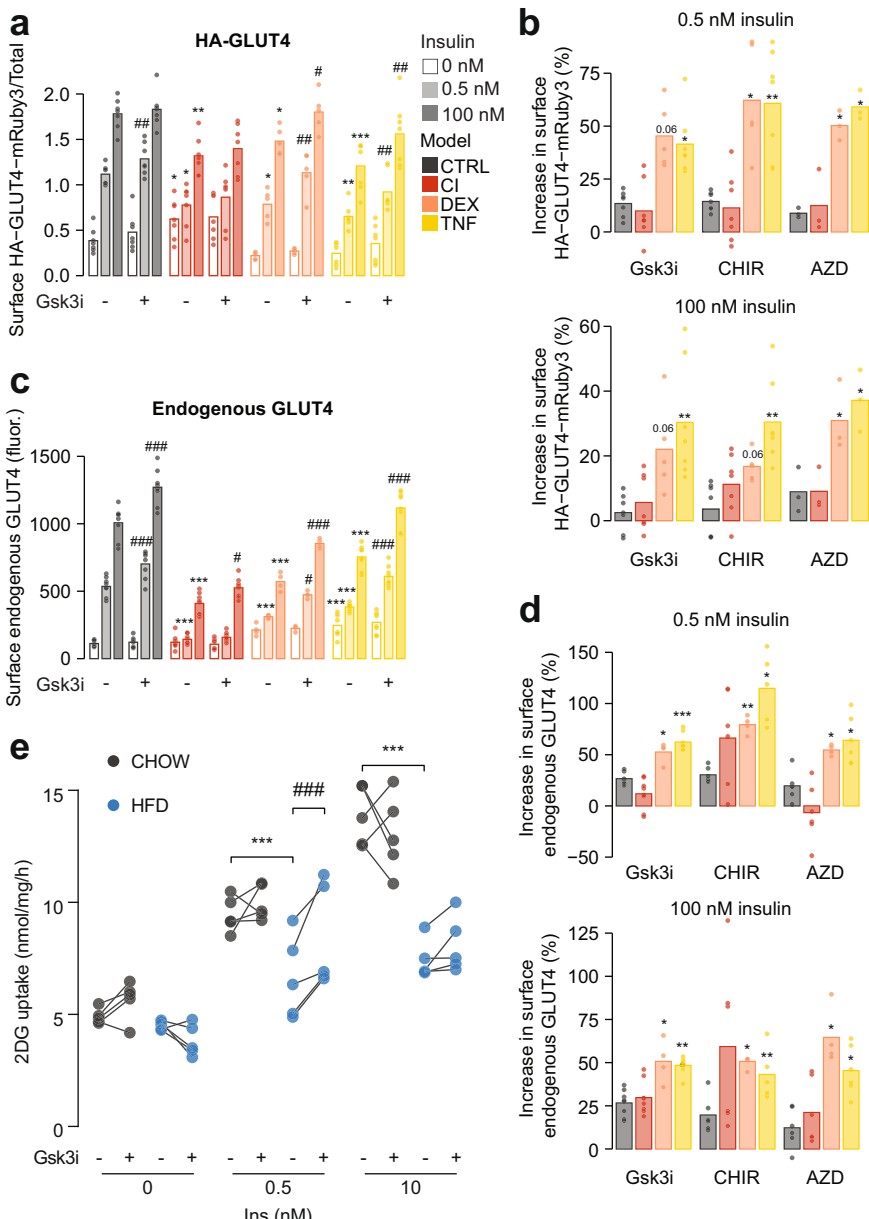

**Fig. 5 | Pharmacological inhibition of GSK3 rescues insulin sensitivity in vitro and ex vivo. a** Control and insulin-resistant 3T3-L1 adipocytes expressing HA-GLUT4-mRuby3 were pretreated for 90 min with DMSO or a 500 nM LY2090314 ('GSK3i') following a 20 min insulin treatment. The normalized abundance of HA-GLUT4-mRuby3 at the plasma membrane was compared between control and insulin-resistant cells treated with DMSO (*), and between GSK3i-treated cells and DMSO-treated cells within each insulin resistance model (#), using two-way ANOVAs and Dunnett's post-hoc tests. *p-values (left-to-right): 0.0164, 0.0191, 0.0080, 0.0359, 0.0190, 0.0011, 0.0003. #p-values (left-to-right): 0.0062, 0.0047, 0.0393, 0.0015, 0.0071. **b** The percentage increase in PM HA-GLUT4 caused by GSK3 inhibition with 500 nM GSK3i, 10 μM CHIR99021 ('CHIR'), or 10 μM AZD2858 ('AZD') at 0.5 nM or 100 nM insulin. Data were analyzed by mixed-effects analysis corrected for multiple comparisons (two-sided Dunnett's test), comparing control vs insulin-resistant models (*). *p-values (left-to-right, top-to-bottom): 0.0576, 0.0127, 0.0186, 0.0038, 0.0284, 0.0147, 0.0572, 0.0063, 0.0566, 0.0022, 0.0377,

0.0403. As in **a**, **b**, using wild-type 3T3-L1 adipocytes and quantification of plasma membrane endogenous GLUT4. *P*-values in **c**, left-to-right *<0.0001, <0.0001, <0.0001, <0.0001, 0.0006, <0.0001, <0.0001. #: 0.0002, <0.0001, 0.0295, 0.0318, <0.0001, <0.0001, <0.0001. *P*-values in **d**, left-to-right, and top-to-bottom: 0.0229, <0.0001, 0.0030, 0.0179, 0.0301, 0.0159, 0.0159, 0.0015, 0.0185, 0.0013, 0.0452, 0.0350. **e** Epididymal fat explants from mice fed CHOW or HFD for 14 d were pretreated with either DMSO or 500 nM LY2090314 ('GSK3i') for 60 min. [3]H-2DG uptake was assessed during a 20 min insulin stimulation. HFD was compared to CHOW in DMSO-treated explants (*), and DMSO-treated explants were compared to GSK3i-treated explants in either HFD or CHOW (#), using two-way repeated-measures ANOVAs followed by Šídák's post-hoc tests. *n* = 5 explants from 5–10 biologically independent animals (pooling details are described in Methods). Lines connect data points from the same mouse. *p-values (left-to-right): 0.0009, <0.0001. #p-value: 0.0008. */#0.01 < *p* < 0.05, **/##0.001 < *p* < 0.01, ***/###*p* < 0.001. Source data are provided as a Source Data file.

## Study limitations

These studies were undertaken exclusively in male mice, and future efforts should establish whether the signaling changes observed also occur in female mice. We also focused exclusively on adipose tissue to leverage the highly insulin-responsive 3T3-L1 adipocyte cell line,

although muscle is a major contributor to whole-body insulin resistance in type 2 diabetes[72]. It will therefore be important to extend our work characterizing insulin resistance-induced signaling changes to muscle. Our studies were restricted to a single time point post insulin-stimulation (10 min)—we reasoned that signaling events most relevant

to glucose uptake should occur by this time, since insulin stimulates GLUT4 translocation within 10 minutes[73], and most acute insulin-induced phosphorylation changes also occur within this timeframe[18]. We cannot exclude, however, the possibility that insulin resistance involves impaired signaling kinetics. Finally, we have focused largely on proteomic and phosphoproteomic changes occurring across multiple cultured and in vivo models of insulin resistance, as we have previously found this approach is successful in identifying causal drivers of metabolic dysfunction[3,22]. However, this does not account for model-specific changes that contribute to insulin resistance. Bona fide changes of this type may contribute meaningfully to insulin resistance within each model, but they are challenging to identify as they cannot be readily separated from changes unrelated to insulin resistance.

## Methods

### 3T3-L1 fibroblast culture and differentiation into adipocytes

Mycoplasma-free 3T3-L1 fibroblasts obtained from 3T3-L1 Howard Green (Harvard Medical School, Boston, MA) were maintained in Dulbecco's Modified Eagle Medium (DMEM) (Thermo Fisher Scientific) supplemented with 10% fetal calf serum (FCS) (Thermo Fisher Scientific) and 1% GlutaMAX (Thermo Fisher Scientific) in a humidified atmosphere with 10% $CO_2$. HA-GLUT4-overexpressing cell lines were generated by retroviral transduction of 3T3-L1 fibroblasts as previously described[74]. Confluent 3T3-L1 cells were differentiated into adipocytes by addition of DMEM/10% FCS/GlutaMAX containing 0.22 μM dexamethasone, 100 ng/mL biotin, 2 μg/mL insulin, 500 μM IBMX (day 0). After 72 h, medium was replaced with DMEM/10% FCS/GlutaMAX containing 2 μg/mL insulin (day three post differentiation). After a further 72 h (day six post differentiation), cells were switched to DMEM/10% FCS/GlutaMAX. Medium was subsequently replaced every 48 h. Cells were used between days 10 and 15 after the initiation of differentiation

### In vitro models of insulin resistance

Insulin resistance was induced by hyperinsulinemia, dexamethasone, tumor necrosis factor-α (TNF), antimycin A, or mitoParaquat (MPQ) as previously described[3,22,23,75]. The chronic insulin (CI) model of hyperinsulinemia was created by incubation of adipocytes with 10 nM insulin for 24 h (add insulin at 1200, 1600, and 2000 h on day 1 and 0800 h the following day, prior to serum starvation at 1200 on day 2). Glucocorticoid-induced insulin resistance was recreated with 20 nM dexamethasone (Dex) (0.01% ethanol carrier as control), starting on day seven post initiation of differentiation and maintained for 8 d. Medium was changed every 48 h. Chronic low-dose inflammation was mimicked in 3T3-L1 adipocytes by incubation with 2 ng/mL TNFα (TNF; Calbiochem) for 4 d. Medium was changed every 24 h. Following CI, Dex of TNF treatment, cells were washed and serum-starved for 1.5 h in the absence of insulin, dexamethasone, or TNFα prior to insulin stimulation. Mitochondrial oxidants were induced by incubation with 50 nM antimycin A or 25 μM mPQ for 1.5 h during serum starvation. Note that 25 μM mPQ was used in this study as all samples for phosphoproteomic analysis were undertaken in day 15 adipocytes, and this dose was required for sufficient induction of mitochondrial oxidants and insulin resistance in these cells.

### 3T3-L1 adipocyte treatment (insulin or kinase inhibitors)

For acute insulin and kinase inhibitor treatments, 3T3-L1 adipocytes were used at 10 days post-differentiation and serum starved for 2 h prior to treatments. For assessment of plasma membrane GLUT4, cells were treated with DMSO, 100 nM LY2090314, 10 μM CHIR99021, 10 μM AZD2858, 0.5 μM PCC020817[76] or 10 μM 39621 for 90 min. For the insulin resistance phosphoproteomics experiment cells were treated with 100 nM insulin or PBS for 10 min. For the GSK3i phosphoproteomics experiments cells were treated with DMSO or 100 nM LY2090314 for 20 min. Cells were then washed three times with ice-

cold TBS before scraping in SDC lysis buffer (4% Sodium Deoxycholate, 100 mM Tris pH 8.5) with heating for 5 min at 95 °C. Lysates were sonicated (75% output power, 2x 30 s) and centrifuged for 20 min at 20,000 x *g* at 0 °C to form a lipid layer. Clarified protein lysate was carefully collected without interfering with the upper lipid layer, and protein concentration was measured using BCA assay.

### Phosphoproteomics sample preparation

240 μg protein was reduced and alkylated in one step using 10 mM Tris (2-carboxyethyl)phosphine (TCEP) and 40 mM 2-Chloroacetamide (CAA) at 45 °C for 5 min. LysC and trypsin (1:100 enzyme-protein ratio) were added to samples to digest the protein in a 96-well plate at 37 °C overnight. 1:6 of the peptides (40 μg) were taken for proteome analysis, and for the remaining phosphopeptides were enriched according to the EasyPhos method[20]. Phosphopeptides were resuspended in 5 μl MS loading buffer (2% ACN, 0.3% TFA) for LC-MS/MS analysis.

### Proteome sample preparation

40 μg peptides were fractionated by offline Strong Cation Exchange using SCX StageTips as described[77]. Briefly, StageTips fabricated with 6-layers of SCX material (3M Empore) were equilibrated sequentially: 1x 100 μL ACN, 1x 100 μL 30% MeOH/1% TFA, 1x 100 μL 0.2% TFA, and de-salted peptides were loaded onto StageTips in 100 μL 1% TFA. Peptides were sequentially eluted with 6 elution buffers of increasing strength as follows: (1) 50 mM $NH_4OAc$/20% ACN/0.5% Formic acid; (2) 75 mM $NH_4OAc$/20% ACN/0.5% Formic acid; (3) 125 mM $NH_4OAc$/20% ACN/0.5% Formic acid; (4) 200 mM $NH_4OAc$/20% ACN/0.5% Formic acid; (5) 300 mM $NH_4OAc$/20% ACN/0.5% Formic acid; (6) 5% ammonium hydroxide/80% ACN. Fractions were collected separately, and dried in a vacuum concentrator (Eppendorf). Peptides were resuspended in MS loading buffer (2% ACN, 0.3% TFA) for LC-MS/MS analysis.

### Liquid chromatography-tandem mass spectrometry (LC-MS/MS)

**DIA-MS (3T3-L1 phosphoproteomes).** Peptides were loaded onto in-house fabricated 50 cm columns with a 75-μM I.D., packed with 1.9 μM C18 ReproSil Pur AQ particles using a Dionex U3000 HPLC coupled to an Orbitrap Exploris 480 mass spectrometer (Thermo Fisher Scientific). Column temperature was maintained at 60 °C with a Sonation column oven, and peptides separated using a binary buffer system comprising 0.1% formic acid (buffer A) and 80% ACN plus 0.1% formic (buffer B), at a flow rate of 400 nl/min, with a gradient of 3–19% buffer B over 80 min followed by 19–41% buffer B over 40 min, resulting in ~2-h gradients. Peptides were analyzed with one full scan (350–1400 m/z, R = 120,000) at a target of 3e6 ions, followed by 48 data-independent acquisition (DIA) MS/MS scans (350–1022 m/z) with higher-energy collisional dissociation (HCD) (target 3e$^6$ ions, max injection time 22 ms, isolation window 14 m/z, 1 m/z window overlap, normalized collision energy (NCE) 25%), with fragments detected in the Orbitrap (R = 15,000).

**DDA-MS (3T3-L1 proteomes).** Peptides were loaded onto 50 cm fused silica columns packed in-house (ReproSil Pur C18-AQ, 1.9 μm particle size) and maintained at 60 °C using a column oven (Sonation, GmbH). Peptides were separated by reversed-phase chromatography using an Easy nLC 1200 (Thermo Fisher Scientific), with a binary buffer system of 0.1% formic acid (buffer A) and 80% ACN/0.1% formic (buffer B). Peptides were separated by linear gradients of buffer B (3 to 25% over 90 min, followed by 25-35% over 20 min, and 35-60% over 10 min) at a flow rate of 300 nL/min, and electrosprayed directly into the mass spectrometer by the application of 2.4 kV with a liquid junction union. Ionized peptides were analyzed using a benchtop Orbitrap (Q Exactive HF-X) mass spectrometer (Thermo Fisher Scientific). The mass spectrometer was operated in data-dependent mode, performing survey scans of 3e$^6$ ions at a resolution of 60,000 from 300–1650 m/z. The 15

most abundant precursors from the survey scan with charge state >1 and <5 were selected for fragmentation. Precursors were isolated with a window of 1.4 m/z and fragmented in the HCD cell with NCE of 27. Maximum ion fill times for the MS/MS scans were 28 ms, with a target of $2.9e^3$ ions (intensity threshold $2.9e^5$ ions). Fragment ions were analyzed with high resolution (15,000) in the Orbitrap mass analyzer. Dynamic exclusion was enabled with duration 30 s.

**DDA-MS (tissue phosphoproteomes).** Peptides were loaded onto 50 cm fused silica columns packed in-house (ReproSil Pur C18-AQ, 1.9 μm particle size) and maintained at 60 °C using a column oven (Sonation, GmbH). Peptides were separated by reversed-phase chromatography using an Easy nLC 1000 coupled to a Q Exactive HF mass spectrometer (Thermo Fisher Scientific). Peptides were separated using a binary buffer system comprising 0.1% formic acid (buffer A) and 80% ACN plus 0.1% formic (buffer B), at a flow rate of 350 nl/min, with a gradient of 3–19% buffer B over 60 min followed by 19–41% buffer B over 30 min Peptides were electrosprayed directly into the mass spectrometer by the application of 2.3 kV with a liquid junction union. The mass spectrometer was operated in data-dependent mode, performing survey scans of $3e^6$ ions at a resolution of 60,000 from 300-1600 m/z, and the 5 most abundant precursors from the survey scan with charge state >1 and <5 were selected for fragmentation. Precursors were isolated with a window of 1.6 m/z and fragmented in the HCD cell with NCE of 25. Maximum ion fill times for the MS/MS scans were 120 ms, with a target of $4e^4$ ions (intensity threshold $3.3e^5$ ions). Fragment ions were analyzed with high resolution (15,000) in the Orbitrap mass analyzer, and dynamic exclusion was enabled with duration 45 s.

**DDA-MS (GSK3i phosphoproteomes).** Peptides were loaded onto 50 cm fused silica columns packed in-house (ReproSil Pur C18-AQ, 1.9 μm particle size) and maintained at 60 °C using a column oven (Sonation, GmbH). Peptides were separated by reversed-phase chromatography using a Dionex U3000 HPLC coupled to a Q Exactive HF-X mass spectrometer (Thermo Fisher Scientific). Peptides were separated using a binary buffer system comprising 0.1% formic acid (buffer A) and 80% ACN plus 0.1% formic (buffer B), at a flow rate of 350 nl/min, with a gradient of 3–19% buffer B over 80 min followed by 19–41% buffer B over 40 min Peptides were electrosprayed directly into the mass spectrometer by the application of 2.4 kV with a liquid junction union. The mass spectrometer was operated in data-dependent mode, performing survey scans of $3e^6$ ions at a resolution of 60,000 from 350–1400 m/z, and the 10 most abundant precursors from the survey scan with charge state >1 and <5 were selected for fragmentation. Precursors were isolated with a window of 1.6/ mz and fragmented in the HCD cell with NCE of 27. Maximum ion fill times for the MS/MS scans were 50 ms, with a target of $2e^4$ ions (intensity threshold $4e^5$ ions). Fragment ions were analyzed with high resolution (15,000) in the Orbitrap mass analyzer, and dynamic exclusion was enabled with duration 30 s.

## MS RAW data processing
**DDA.** RAW data was analyzed using MaxQuant[78] (v1.6.0.9, v1.6.1.0 and v1.6.17.0 for the adipose tissue phosphoproteomes, 3T3-L1 proteomes and GSK3i phosphoproteomes, respectively), searching against the *Mus musculus* UniProt database (July 2017 and December 2019 releases for the tissue phosphoproteome and 3T3-L1 proteome, or GSK3i phosphoproteome). Default settings were used, with the addition of "Phospho(STY)" as a variable modification, and "match between runs" was turned on for all samples analyzed in the same runs.

**DIA.** RAW data was analyzed using Spectronaut (v14.11.210528.47784). Data were searched using directDIA against the *Mus musculus* UniProt Reference Proteome database (January 2021 release), Precursor and protein Qvalue cutoffs 0.01, Qvalue filtering, MS2 quantification, and

"PTM localization" switched on. PTM "Probability cutoff" was set to 0 and localization filtering was performed during downstream analysis. Spectronaut output tables were processed using the Peptide collapse (v1.4.2) plugin for Perseus[21,79].

## 2DG uptake assays in 3T3-L1 adipocytes
Following 1.5 h serum-starvation in DMEM/0.2% BSA/1% GlutaMAX, cells were washed and incubated in pre-warmed Krebs–Ringer phosphate buffer containing 0.2% bovine serum albumin (BSA, Bovostar, Bovogen) (KRP buffer; 0.6 mM $Na_2HPO_4$, 0.4 mM $NaH_2PO_4$, 120 mM NaCl, 6 mM KCl, 1 mM $CaCl_2$, 1.2 mM $MgSO_4$ and 12.5 mM Hepes (pH 7.4)). Cells were stimulated with the indicated dose of insulin (typically 100 nM insulin) for 20 min or as indicated. To determine non-specific glucose uptake, 25 μM cytochalasin B (ethanol, Sigma Aldrich) was added to the wells before addition of 2-[3H] deoxyglucose (2DG) (PerkinElmer). During the final 5 min 2DG (0.25 μCi, 50 μM) was added to cells to measure steady-state rates of 2DG uptake. Following three washes with ice-cold PBS, cells were solubilised in PBS containing 1% (v/v) Triton X-100. Tracer uptake was quantified by liquid scintillation counting and data normalized for protein content.

## Assessment of plasma membrane GLUT4
Plasma membrane GLUT4 was determined in 3T3-L1 adipocytes expressing the GLUT4 reporters HA-GLUT4[74] (Supplementary Fig. 4l) or HA-GLUT4-mRuby (Fig. 5a, b, Supplementary Fig. 7b, c), or in wildtype 3T3-L1 (Fig. 5c, d, Supplementary Fig. 7d, e). Cells were serum-starved for 2 h in DMEM/0.2% BSA/GlutaMAX in a CO2 incubator. Cells were stimulated with 0.5 or 100 nM insulin (as indicated) for indicated times. Cells were fixed in 4% PFA but not permeabilized. Residual PFA was quenched with 50 mM Glycine in PBS, followed by blocking in 5% Normal Swine Serum (NSS; Jackson ImmunoResearch) for 30 min at room temperature. The amount of HA-GLUT4 or HA-GLUT4-mRuby3 present at the plasma membrane was determined by the accessibility of the HA epitope to anti-HA antibody (Covance, clone 16B12). For wild-type cells, GLUT4 present at the plasma membrane was labeled with human anti-GLUT4 antibody (LM048[60]; kindly provided by Joseph Rucker, Integral Molecular, PA, USA) that recognises an exofacial epitope in GLUT4. Cells were incubated with 20 μg/mL Alexa-488-conjugated anti-mouse IgG antibody (for anti-HA staining) or Alexa-488-conjugated anti-humanIgG antibody (for anti-GLUT4 staining). Antibody incubations were performed in 2% NSS in PBS for 1 h at room temperature.

**HA-GLUT4 cells.** Determination of total HA-GLUT4 was performed in a separate set of cells that underwent the same labeling procedure except that anti-HA staining was performed after permeabilization of the cells with 0.1% (w/v) saponin. Total HA-GLUT4 was measured separately for each experimental treatment group. Cells were stored in PBS containing 2.5% DABCO, 10% glycerol, pH 8.5. Fluorescence (excitation 485 nm/emission 520 nm) was measured using a fluorescent microtiter plate reader (FLUOstar Galaxy, BMG LABTECH). Surface HA-GLUT4 was expressed as a percentage of total HA-GLUT4.

**HA-GLUT4-mRuby3 and wild-type cells.** Secondary antibody incubations also included Hoechst 33342 (Life Technologies) to label nuclei. Cells were stored in PBS containing 2.5% DABCO, 10% glycerol, pH 8.5, and imaged using the Opera Phenix High Content Screening System (Perkin Elmer). Confocal images (basal section for wild-type cells, mid-section for HA-GLUT4-mRuby cells) were obtained using a 20x water objective (N.A 1.0), with 2-pixel binning, 9 images per well. Excitation wavelengths and emission filters used were as follows: endogenous surface GLUT4 and HA-tagged surface GLUT4: 488 nm, 500-550 nm; mRuby: 561 nm, 570-630 nm; Hoechst: 405 nm, 435-480 nm. Images were

analyzed using Harmony phenoLOGIC Software (v4.9; Perkin Elmer) to quantify cell-associated anti-HA (Alexa-488), mRuby3 (as a measure of total cellular GLUT4) or anti-GLUT4 (Alexa-488) fluorescent signals. Surface HA-GLUT4-mRuby3 was expressed relative to the total mRuby3 signal. Surface endogenous GLUT4 was expressed as raw Alexa-488 fluorescence.

## Protein knockdown by siRNA transfection

siRNA-based protein knockdown was performed by reverse transfection of 3T3-L1 adipocytes (d6–d7 post differentiation) as previously described[80]. Briefly, Opti-MEM (Thermo Fisher Scientific) and TransIT-X2 ® (Mirus, MIR6006) were mixed (ratio 30/1) and kept at room temperature for 20 min. siRNA was added to the Opti-MEM/TransIT-X2 mixture to a final concentration of 50 nM/well, mixed gently by pipetting and incubated at room temperature for 30 min. Transfection reagents were transferred into either 96-well plates (for plasma membrane GLUT4 assay) or 24 well-plates (for RNA extraction) pre-coated with matrigel (1:100). Cells were reseeded onto transfection reagents. Media was replaced 24 h after reseeding (DMEM/10%FBS/GlutaMAX) and every 48 h for 5 days. Following assessment of plasma membrane GLUT4, normalization was performed by linearly transforming GLUT4 fluorescence from each experiment to set the 0 nM insulin NT value to 0 and the 100 nM insulin NT value to 1.

## RT-qPCR

Total RNA was extracted from tissues using TRIzol reagent (Invitrogen) as per the manufacturer's protocol except that 1-bromo-3-chloropane (Sigma) was used in place of chloroform for phase separation. RNA was precipitated using isopropanol, washed using 70% ethanol and reconstituted in DEPC water. RNA quality and quantity was determined by using a Nanodrop 2000. 500 ng of RNA was reverse transcribed into cDNA using PrimeScript Reverse Transcriptase (Clontech, Takara Bio Company) as per the manufacturer's protocol using oligo-dT primers. PCR reactions were carried out using 1 in 10 diluted cDNA and SYBR Select Master Mix (Thermo-Fisher Scientific) on the LightCycler 480 II (Roche) with the following primer sets; Mark2 F 5′ CCTCCAAGCTTCTCCATTCCC 3′ and Mark2 R 5′ AATCAAGGTGTCCCAAGGTGG 3′ or Mark3 F 5′TGTTCGAAGTC ATTGAAACGGAA 3′ and Mark3 R 5′ CCTTCATTCTTCCATGTGCAACC 3′. All samples were normalised to a housekeeping gene CyclophilinB with the following primer pair; CyclophilinB F 5′ TTCTTCATAACCAC AGTCAAGACC 3′and CyclophilinB R 5′ ACCTTCCGTACCACATCC AT3′. Assay efficiencies were checked using serial dilution of a pooled control sample, and only primers that fell in the 85–120% range were used.

## Phosphatase inhibitor treatment of 3T3-L1 adipocytes

3T3-L1 adipocytes were serum-starved for 1.5 h and treated with or without calyculin A (50 nM, DMSO) (Cell Signaling Technologies) or okadaic acid (1 μM, H$_2$O) (Merck, Sigma) for 10 or 20 min in the absence of presence of insulin as indicated. In all cases, the inhibitor was added at the same time and for the same duration as insulin. Vehicle controls were used as appropriate for each inhibitor.

## In vitro serine/threonine protein phosphatase activity assay

Control 3T3-L1 adipocytes or adipocytes treated with insulin (CI), dexamethasone (Dex) or TNFα (TNF) to induce insulin resistance were serum started for 1.5 h and left unstimulated or treated with 100 nM insulin for 10 min Cells were then washed thrice with cold PBS and scraped in cold lysis buffer (20 mM Tris-HCl (pH 7.5), 1 mM Na-EGTA, 1% NP40 containing EDTA-free protease inhibitors (Roche)). Lysates were needle-homogenized and the fat cake removed by centrifugation. To assess global serine/threonine protein phosphatase activity, lysates were diluted up to 50 μL in lysis buffer before the addition of 50 μL of phosphatase assay buffer (20 mM Tris-HCl, pH 7.5, 5 mM NiCl

(required for protein phosphatase 2B activity), 10 mM MgCl$_2$, 2 mM MnCl$_2$ (required for protein phosphatase 1 activity), 10 mM CaCl$_2$ (required for protein phosphatase 2B activity) and 10 mM pNPP). Reactions were incubated for 30 min at 37 °C before being neutralized by the addition of 50 μL of 2 M NaOH and absorption at 405 nm immediately measured (Tecan Infinite M1000). A405 were blanked to reactions containing naïve lysis buffer. Phosphatase activity was calculated by determining the amount of pNPP converted to NPP, using the molar extinction coefficient for pNPP ($18,000\ M^{-1}\ cm^{-1}$), and normalizing to the amount of protein and reaction time. Different amounts of control lysate were included in each run to confirm linearity and control lysate was incubated with a phosphatase inhibitor cocktail (final concentrations: 2 mM sodium orthovanadate, 1 mM sodium pyrophosphate, 10 mM sodium fluoride) as a control for assay specificity.

## Western blotting

**Sample preparation.** To assess GSK3 phosphorylation in insulin resistance models, protein was extracted as described above (*3T3-L1 adipocyte treatment*). Chloroform-methanol precipitation was performed as previously described[81] with an initial lysate:chloroform:methanol ratio of 1:1:4. and protein was reconstituted in 2% SDS following quantification by BCA assay (Thermo Fisher Scientific) and dilution in Laemmli buffer. To assess the efficacy of GSK3 and MARK inhibitors, cells were incubated with DMSO or the specified dose of inhibitor for 2 h in serum-free media. Cells were washed three times in ice-cold PBS and lysed in RIPA buffer containing protease and phosphatase Inhibitors (Thermo Fisher Scientific). Lysates were then sonicated and centrifuged at 16,000 x *g* at 4 °C for 30 min Protein concentration of the supernatant was quantified by BCA Assay (Thermo Scientific) following dilution in Laemmli buffer.

**SDS-PAGE and immunoblotting.** 10-20 μg of protein was resolved by SDS-PAGE, transferred onto PVDF membranes (GSK3 phosphorylation, MilliporeSigma) or nitrocellulose membranes (GSK3/MARK inhbitors, Bio-Rad) and immunoblotted as previously described[57]. Primary antibodies used were pS21/S9 GSK3α/β (9331S, Cell Signaling Technology), GSK3β (9315, Cell Signaling Technology), 14-3-3 (SC629, Santa Cruz Biotechnology), α-tubulin (T9026 Sigma Aldrich), pS641 glycogen synthase (47043, Cell Signaling Technology), glycogen synthase (3886, Cell Signaling Technology) and pS246/S259/S155 HDAC4/5/7 (3443, Cell Signaling Technology).

Membranes were incubated with the appropriate HRP-conjugated or Alexa Fluor-conjugated secondary antibodies for 1-2 h at room temperature. For GSK3 phosphorylation experiments protein bands were visualized by ECL (MilliporeSigma) on a LI-COR C-DiGit blot scanner (LI-COR Biosciences) or by 800-fluorescence intensity on a LI-COR Odyssey CLx imager. For GSK3/MARK inhibitor experiments protein bands were visualized using ECL (Thermo Fisher Scientific) or 647-fluorescence intensity on the Chemidoc MP (Bio-Rad).

## Animal details

Eight-week-old male C57BL/6J mice were purchased from Australian BioResources (Moss Vale, Australia). The animals were kept in a temperature-controlled environment (22 ± 1 °C, humidity 44-46%) on a 12 h light/dark cycle with free access to food and water. Following two weeks acclimatization (10 week old), mice were split into three groups: 1) mice fed with standard lab diet (CHOW) (13% calories from fat, 22% calories from protein, and 65% calories from carbohydrate, 3.1 kcal/g; Gordon's Specialty Stock Feeds, Yanderra, Australia)for 14 d; 2) mice fed a high fat high sucrose diet (HFD; 47% of calories from fat (40% calories from lard), 21% calories from protein, and 32% calories from carbohydrates (16% calories from starch), 4.7 kcal/g) for 14 d; and 3) mice fed an HFD diet for 14 d, before returning to CHOW for 5 d. All experiments were carried out with the approval of the University of

Sydney Animal Ethics Committee (2014/694), following guidelines issued by the National Health and Medical Research Council of Australia. All studies used at least 24 mice per treatment group ($n$ = 12 saline-treated, $n$ = 12 insulin-stimulated), and studies were performed over 4 separate days spanning two weeks. Due to low adipose tissue weight in chow-fed mice, lysates from adipose tissue from two mice were combined in order to obtain enough starting material for phosphoproteomics.

## Assessment of body composition

Body composition of individual mice was assessed using the Echo-MRI 900 to determine lean mass, 24 h prior to euthanasia. Analysis was performed as per the manufacturer's specifications.

## In vivo insulin stimulation

Mice were fasted from 12:00 and experiments were carried out between 18:00 and 21:00 so that mice received insulin approximately in line with their natural circadian cycle of insulin release[82,83]. Administration of saline/insulin and 2DG tracer via the hepatic portal vein was performed as previously described[84] with the following minor modifications. Mice were injected with 80 mg/kg lean mass (determined by Echo-MRI) pentobarbitone intraperitoneally. Following induction of anesthesia (approximately 15–20 min after injection), mice were placed on a heat pad (~30 °C), and the hepatic portal vein accessed via an incision to access the abdominal cavity. A bolus of saline or insulin (1 U/kg lean body mass) containing 5–10 µCi [3H]2DG tracer was administered via the hepatic portal vein. Blood samples were taken via the tail vein throughout the procedure to assess blood glucose concentrations and 2DG tracer. After 10 min, epididymal adipose tissue was excised and rapidly rinsed in ice-cold PBS to remove non-adipose material such as blood. Ice-cold PBS was used to reduce kinase and phosphatase activity, hence minimizing phosphoproteome changes. Adipose tissue was then immediately snap-frozen in liquid nitrogen. Mice were euthanized by cervical dislocation.

## Adipose tissue lysis

Snap-frozen epididymal adipose tissue was heated for 5 min at 95 °C with shaking in SDC lysis buffer (4% Sodium Deoxycholate, 100 mM Tris pH 8.5). Lysates were sonicated (75% output power, 1x 15 s) and heated for an additional 5 min at 95 °C with shaking. Lysate was centrifuged for 20 min at 21,000 x $g$ at 0 °C to form a lipid layer. Clarified protein lysate was carefully collected without interfering with the upper lipid layer, and protein concentration was measured using BCA assay.

## Adipose tissue lysate processing to determine 2DG uptake

Aliquots of lysed tissue in SDC buffer were treated with an equal volume of 1% trifluoroacetic acid to precipitate SDC and samples were centrifuged. Cleared lysates were transferred to a 96-well plate and dried using a vacuum centrifuge. Samples were resuspended in 10mM Tris-HCL pH8.5. Samples were transferred onto columns containing AG1-X8 resin to trap phosphorylated 2DG (2-[3H] DG-6-P). Resin was washed three times with ddH$_2$O, before 2-[3H] DG-6-P was eluted with 2M NaCl, 1% trifluoroacetic acid. Eluants were assessed for radioactivity by liquid scintillation counting. Tracer in the blood was measured at 0, 1, and 10 min during the experiment, and the average tracer count (DPM) between 1 and 10 min used to calculate the specific activity (DPM/mol glucose) and the glucose transport index[84]. Uptake data were normalized to tissue protein content.

## 2DG uptake assays in adipose tissue explants

Assessment of insulin-stimulated 2DG uptake in adipose explants was performed as previously described[85] with minor modifications.

Epididymal fat depots were excised from mice, transferred immediately to warm basal medium (DMEM/20 mM HEPES/2% bovine serum albumin (BSA, Bovostar, Bovogen), pH 7.4), and minced into fine pieces. Explants were serum-starved for 2 h. The [3H]2DG uptake assay was performed in Krebs–Ringer phosphate buffer containing 2% BSA (KRP buffer; 0.6 mM Na$_2$HPO$_4$, 0.4 mM NaH$_2$PO$_4$, 120 mM NaCl, 6 mM KCl, 1 mM CaCl$_2$, 1.2 mM MgSO$_4$ and 12.5 mM HEPES (pH 7.4)), whereby insulin and/or calyculin A (100 nM) was added for 20 min, and glucose transport was initiated by addition of [3H]2DG (0.25 µCi, 50 µM) and [14C]mannitol (Perkin Elmer, 0.036 µCi/sample) for the final 5 min of the assay to measure steady-state rates of 2DG uptake. The equivalent volume of vehicle (DMSO) was included for cells not treated with Calyculin A. Samples were assessed for radioactivity by scintillation counting and [14C]mannitol was used to correct for extracellular [3H] 2DG. Fat explants were lysed in 100 nM NaOH for protein determination and assay normalization to total protein.

## Inhibition of GSK3 in adipose tissue explants

Fifteen 18-week old male C57Bl/6J were fed a standard lab diet (CHOW; 13% calories from fat, 22% calories from protein, and 65% calories from carbohydrate, 3.1 kcal/g; Gordon's Specialty Stock Feeds, Yanderra, Australia; $n$ = 10 mice) or high-fat high-sucrose diet (HFD; 45.5% of calories from fat (40% calories from lard), 20.5% calories from protein, and 34% calories from carbohydrate (14% calories from starch), 4.95 kcal/g; $n$ = 5 mice) for 14 d. CHOW mice were paired and their epididymal fat depots were combined to obtain sufficient starting material. Fat depots were then minced and divided into two explants per pooled mouse. After washing and 2 h incubation in DMEM/2% BSA/20 mM HEPES, pH 7.4, each pair of explants was exposed to 0.005 % (v/v) DMSO with or without 500 nM LY2090314 for 1 h. The 2DG uptake assay was then carried out as described above, allowing the assessment of the effects of both insulin and LY2090314 within each mouse by two-way repeated-measures ANOVA and Šídák's post-hoc tests using GraphPad Prism 9.3.1.

## Phosphoproteome and proteome data processing and analysis

Phosphoproteomics and proteomics data were analyzed in the R programming environment using RStudio (R version: 4.0.3, RStudio version: 1.3.1093-1). Most visualizations were generated using the R package "ggplot2" (version 3.4.0).

**Replicates.** The number of biological replicates per study are as follows: 3T3-L1 insulin resistance (IR) phosphoproteomics: CTRL unstimulated (BAS) $n$ = 6; CI, DEX, TNF, MPQ, and AA BAS $n$ = 4; CTRL insulin stimulated (INS) $n$ = 6; CI, DEX, TNF, MPQ, and AA INS $n$ = 4. 3T3-L1 IR proteomics: CTRL BAS $n$ = 5, CI BAS $n$ = 3, DEX BAS $n$ = 4, TNF BAS $n$ = 3, MPQ BAS $n$ = 4, AA BAS $n$ = 4, CTRL INS $n$ = 5, CI INS $n$ = 4, DEX INS $n$ = 4, TNF INS $n$ = 4, MPQ INS $n$ = 4, AA INS $n$ = 4. Mouse phosphoproteomics: CHOW BAS $n$ = 12, HFD BAS $n$ = 12, REV BAS $n$ = 12, CHOW INS $n$ = 12, HFD INS $n$ = 12, REV INS $n$ = 12. GSK3i phosphoproteomics: Basal $n$ = 4, GSK3i $n$ = 4.

**Filtering, imputation, normalization.** Phosphopeptides/proteins that were quantified in fewer than eight replicates in the 3T3-L1 IR proteomics and mouse phosphoproteomics studies were removed. Additionally, proteins that were not quantified in any CTRL replicates were removed from the 3T3-L1 IR proteomics study. Following filtering, LFQ intensities were log2-transformed and median normalized. In the 3T3-L1 and mouse IR phosphoproteomics studies, imputation was first performed within each model/diet including CTRL and CHOW. Briefly, if phosphopeptide X was poorly quantified in Basal (BAS) and well quantified in Insulin (INS) for model Y, or vice versa, all missing values in BAS were imputed by sampling from a downshifted normal distribution: N(val$_{min}$ −1, sigma$^2_{wellquant}$), where val$_{min}$ is the minimum intensity for phosphopeptide X across all conditions, and

sigma$_{wellquant}$ is the standard deviation for phosphopeptide X within INS replicates for model Y. The same technique was applied to the GSK3i phosphoproteome study, using Basal and GSK3i-treated instead of BAS and INS. The criteria for "poorly quantified" and "well quantified" were respectively as follows. 3T3-L1 phosphoproteomics: 0 quantifications, >= 5/6 quantifications (CTRL), or 4/4 quantifications (CI, DEX, TNF, MPQ, AA). Mouse phosphoproteomics: 0/12 quantifications, >= 7/12 quantifications. GSK3i phosphoproteomics: <= 1/4 quantification, >= 3/4 quantifications. For GSK3i phosphoproteomics, in the case that only 1 value was quantified in BAS but was higher than the mean of GSK3i values (or vice versa), imputation was not performed. A second step of imputation was performed on the mouse and GSK3i phosphoproteomics studies, where if a phosphopeptide was quantified at least five (mouse) or three (GSK3i) times in a given diet and treatment, the remaining missing values were imputed by sampling from a normal distribution: $N(mu_{self}, sigma^2_{self})$, where $mu_{self}$ and sigma$_{self}$ are the mean and standard deviation within that diet and treatment, for that phosphopeptide.

## Statistical analysis

**3T3-L1 IR phosphoproteomics.** To characterize the response to insulin in insulin sensitive cells, *t*-tests were performed (base R function "t.test") comparing CTRL BAS to CTRL INS for all phosphopeptides that were quantified at least twice in both conditions. *p*-values were adjusted by the Benjamini-Hochberg procedure. Next, CTRL INS values were normalized to CTRL BAS by subtracting the CTRL BAS median. This was repeated in each insulin resistance model, and the resulting insulin responses were compared by ANOVA (base R function "aov") for all phosphopeptides quantified at least twice in BAS and INS for both CTRL and one or more models. For phosphopeptides that were significant in the ANOVA after p-value adjustment, Dunnett's post-hoc tests ("glht" function from "multcomp" version: 1.4−16) were performed to compare the insulin response in each model to that of CTRL cells, and the resulting *p*-values were adjusted within each model.

**3T3-L1 IR proteomics.** First, BAS and INS values were compared within each group by *t*-tests, followed by Benjamini-Hochberg adjustment. BAS and INS values were then pooled within each group, and groups were compared by ANOVA. For proteins that were significant in the ANOVA after Benjamini-Hochberg p-value adjustment, Dunnett's post-hoc tests were performed to compare each insulin resistance model to CTRL cells, and the resulting p-values were adjusted within each model.

**Mouse adipose tissue phosphoproteomics.** To characterize the response to insulin within each diet, BAS and INS values were compared using empirical Bayes moderated *t*-tests implemented in the R package "Limma" (version: 3.14)[86,87]. *p*-values were adjusted within each diet by the Benjamini-Hochberg procedure.

**GSK3i phosphoproteomics.** To assess the effect of GSK3 inhibition, Basal and GSK3i values were compared by standard t-tests for all phosphopeptides quantified at least twice in each condition, followed by Benjamini-Hochberg p-value adjustment.

**Insulin-regulated, defective, and emergent phosphopeptides.** In the 3T3-L1 IR and mouse phosphoproteomics studies, insulin-regulated phosphopeptides were defined as those where the difference between BAS and INS in CTRL or CHOW was significant (adj. *p* < 0.05) and of sufficient magnitude (log2 INS/BAS > 0.58 corresponding to INS/BAS > 1.5, or log2 INS/BAS < −0.58 corresponding to INS/BAS < 0.67).

Defective phosphopeptides were defined as those that responded to insulin under insulin sensitive conditions but failed to respond to

insulin in insulin resistance. In the 3T3-L1 IR phosphoproteomics study, an insulin-regulated phosphopeptide was considered defective in a given model if its insulin response in that model was significantly different to CTRL (adj. *p* < 0.05) and sufficiently small (log2 INS/BAS < 0.58 for CTRL up-regulated phosphopeptides, log2 INS/BAS > −0.58 for CTRL down-regulated phosphopeptides). In the mouse phosphoproteomics study, a phosphopeptide was considered defective in HFD if it was insulin-regulated in CHOW but not HFD (HFD log2 INS/BAS < 0.58 for CHOW up-regulated phosphopeptides, log2 INS/BAS > −0.58 for CHOW down-regulated phosphopeptides).

Emergent phosphopeptides were defined as those that did not respond to insulin under insulin sensitive conditions but did respond in insulin resistance. In the 3T3-L1 IR phosphoproteomics study, a phosphopeptide was considered emergent in a given model if the insulin response in that model was of sufficient magnitude (log2 INS/BAS > 0.58 or < −0.58) and significantly different to CTRL (adj. *p* < 0.05), and if the phosphopeptide did not respond to insulin in CTRL (adj. *p* >= 0.05, −0.58 <= log2 INS/BAS <= 0.58). In the mouse phosphoproteomics study, a phosphopeptide was considered emergent if it was regulated by insulin in HFD (adj. p < 0.05, log2 INS/BAS > 0.58 or < −0.58) but not in CHOW (adj. *p* >= 0.05, −0.58 <= log2 INS/ BAS <= 0.58).

In the 3T3-L1 IR phosphoproteomics study, defective phosphopeptides were further analyzed to attribute defects to changes in the basal or insulin-stimulated states. First, two ANOVAs were performed for each defective phosphopeptide using either BAS or INS values, comparing only CTRL and the specific models that were defective. Dunnett's post-hoc tests were performed on phosphopeptides that were significant in the ANOVA after Benjamini-Hochberg p-value adjustment, and the resulting Dunnett's p-values were adjusted within each model. For a given phosphopeptide X that was up-regulated by insulin in CTRL and defective in model Y, an insulin defect was declared if the difference in INS values between model Y and CTRL was significant (adj. p < 0.05), of sufficient magnitude (FC > 1.5), and changed in the correct direction (model Y < CTRL). Similar filtering was performed for down-regulated phosphopeptides and for identifying basal defects.

**Kinase-based analysis.** Mouse kinase-substrate annotations were extracted from PhositePlus([28], version: Dec 2, 2019) and mapped into the 3T3-L1 IR phosphoproteome by matching uniprots and phosphosite positions. Annotations for kinase isoforms were merged, for example, Akt1, Akt2, and Akt3 were merged, and GSK3a and GSK3b were merged. Kinase substrate enrichment analysis (KSEA) was then performed with the "ksea_batchKinases" function from the R package "ksea"[27](version: 0.1.2), using the log2 INS/BAS fold changes from each model. Kinases with fewer than 10 substrates quantified in each model were considered irrelevant and were excluded. 1000 permutations were performed to determine empirical *p*-values.

**Pathway enrichment.** Mouse Gene Ontology (GO) Biological Processes and Cellular Compartments were extracted using the R packages "org.Mm.eg.db" ([88], version 3.15.0) and KEGG canonical pathways were downloaded from the MSigDB collections ([89,90], version 7.2). Pathway enrichment was performed by either one-sided Fisher's exact test or gene set test using the "geneSetTest" function from the R package "Limma" (version: 3.14)[87]. In the case of gene set test, log2 Model/CTRL fold changes were provided as the enrichment statistic and 9999 random samples were used to determine empirical *p*-values. In order to separately identify up-regulated and down-regulated pathways separate tests were performed using "up" and "down" as the alternative hypothesis.

Functional protein-protein interaction networks were constructed and analyzed using the STRING web app (https://string-db.org/,

version 11.0b). First, two networks were constructed using all proteins either up-regulated or down-regulated in two or more models compared to CTRL cells. Interaction scores were calculated using the Experimental, Database, and Coexpression evidence channels as these were considered the most relevant, and only high-confidence interactions were retained (score > 0.7). Markov clustering was then performed using the default inflation parameter value of 3, and clusters containing ten or more proteins were retained for further analysis. To functionally characterize these clusters pathway enrichment was performed in STRING using the KEGG and Reactome pathways.

**GSK3i substrate identification.** In the GSK3i phosphoproteomics study, potential GSK3 substrate phosphopeptides were defined as those that were decreased by GSK3 inhibition (adj. $p$ < 0.05, log2 GSK3i/Basal < −0.58) and matched the GSK3 substrate motif. To phosphorylate a serine/threonine GSK3 ordinarily requires a priming phosphoserine/threonine four residues downstream, however some GSK3 substrates have a gap of three[91] or five residues[54], implying the general substrate motif pS/T X(2-4) *pS/T, where * indicates the priming phosphosite. Hence a given phosphoserine/threonine matched the GSK3 motif if a priming phosphoserine/threonine was identified 3-5 residues downstream in our phosphoproteomics data or in the PhosphositePlus database ([28], version: Jan 27, 2021).

**Canonical insulin signaling proteins.** Canonical insulin proteins were compiled as previously described[92], compiling annotations from Gene Ontology (GO), Kyoto Encyclopedia of Genes and Genomes, Reactome, and our previous publication[18].

### Reporting summary
Further information on research design is available in the Nature Portfolio Reporting Summary linked to this article.

## Data availability
Source data are provided in this paper. RAW data were processed with MaxQuant using UniProt reference databases (adipose tissue phosphoproteome (July 2017), 3T3-L1 proteome and GSK3i phosphoproteome (December 2019), 3T3-L1 phosphoproteome (January 2021)). All RAW and processed MS data have been deposited in the PRIDE proteomeXchange repository and are accessible with the accession PXD032913. Processed data are available as supplementary tables and can be explored online at www.adipocyteatlas.org. Source data are provided in this paper.

## Code availability
All code used to analyze data and produce figures has been uploaded to https://github.com/JulianvanGerwen/IRAdipPhos.

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

## Acknowledgements

This work was funded by a UKRI grant (MR/S007091/1) to D.J.F. and by NHMRC Project Grants GNT1120201 and GNT1061122 to D.E.J. D.E.J. is an Australian Research Council (ARC) Laureate Fellow. D.J.F. was supported by a Medical Research Council Career Development Award (MR/S007091/1) and a Wellcome Institution Strategic Support Fund award (204845/Z/16/Z). The content is solely the responsibility of the authors and does not necessarily represent the official views of the NHMRC or ARC. The authors acknowledge the facilities, and the scientific and technical assistance of the Sydney Mass Spectrometry Facility, the Sydney Preclinical Imaging Facility and the Laboratory Animal Services at the Charles Perkins Centre, University of Sydney. These studies were supported by the Wellcome-MRC, Institute of Metabolic Science, Metabolic Research Laboratories, Imaging Core (Wellcome Trust Major Award [208363/Z/17/Z]). We thank Marco Dupuis-Rodriguez and Emily Naden for technical assistance. For the purpose of open access, the author has applied a CC-BY public copyright licence to any Author Accepted Manuscript version arising.

## Author contributions

Conceptualization: D.J.F, P.Y., D.E.J., S.J.H. Investigation: D.J.F., J.vG., K.C.C., X.D., A.D.V., S.M, D.M.N., A.S.S., J.R.K., M.R.W., J.T.B., S.J.H. Methodology: D.J.F., J.vG., K.C.C., J.R.K., J.G.B., S.J.H. Formal analysis: D.J.F., J.vG., E.J.N., P.Y., S.J.H. Visualization: D.J.F., J.vG., S.J.H. Software: J.vG., S.J.H. Resources: D.J.F., M.R.W., J.T.B., D.E.J. Project administration: D.J.F., J.vG., S.J.H. Funding acquisition: D.J.F., D.E.J. Supervision: D.J.F., E.J.N., D.E.J., S.J.H. Writing—original draft: D.J.F., J.vG., D.E.J., S.J.H. Writing—review and editing: All authors.

## Competing interests

The authors state that they have potential conflicts of interest regarding this work: M.R.W. and J.T.B. were employees of Eli Lilly during the study. The remaining authors declare no other competing interests.
