## [Peer Review File · Nature Communications]

Phosphoproteomics reveals rewiring of the insulin signaling network and multi-nodal defects in insulin resistanceREVIEWER COMMENTS

Reviewer #1 (Remarks to the Author):

In this study, Fazakerley and colleagues focused on MS analysis of phosphoproteomics to investigate changes in insulin signaling networks in adipocytes using both cell and animal models. They present a large number of datasets mostly based on comparative phosphoproteomics. There are also experiments to look at global protein expression in adipocytes in both model cells and animals. Overall authors have indeed presented greatly improved view of insulin signaling and development of insulin resistance using multiple insulin resistance model.

Using three well characterized models mimicking systemic insulin insults that induce insulin resistance (CI, DEX, and TNF-alpha) plus two models of acute mitochondrial oxidants (MPQ and AA), authors identified both attenuated phosphorylation and emergence of phosphorylation that were insulin responsive and insulin-regulated. Despite observation of substantial requiring of insulin signaling across models, the regulation of canonical insulin-responsive kinases, proteins and phosphoproteins was overwhelmingly unchanged. Using KESA pipeline, they identified hundreds of GSK3 substrates which pointed to dysregulation of GSK3 activity. Normalizing phosphorylated GSK3 to the total protein abundance, they suggested that decreased phosphorylation of GSK3 in S21 or S9 was due to reduced overall abundances of GSK3 and also decreased AKT signaling. They also reported that insulin sensitivity was impaired in the adipose tissue of a high fat high sucrose mice, but this impairment was partially reversed when mice fed HFD for 14 days were switched back to CHOW diet. They tested GSK3 deactivation by insulin by treating with GSK3 inhibitors, and observed GLUT4 in all models but the response was greater for TNF-alpha and DEX models but they were observed only under insulin-stimulation.

The manuscript presents a very elegant comparative phosphoproteomics that critically examines the impact of insulin resistance in diverse cell and animal models. The study examined the phosphoproteome and then incorporated the total proteome to allow for normalization of phosphoproteome to the level of site-specific phosphorylation or phosphosite stoichiometry. These data were followed by extensive pathways mapping and other bioinformatics analysis to pinpoint changes at pathway levels. Overall, this is a well-written manuscript with interesting and complex experimental design and contains information that can improve our level of understandings about dysregulated insulin signaling and development of insulin resistance in different insulin-resistance models. Authors argue that impaired decreases in phosphorylation is a defining feature of insulin resistance.

However, at the end, my enthusiasm was dampened by the lack of validation or follow-up studies to obtain better biological insights that may preclude its publication in Nature Communication.

Comments:

1) Their phosphoproteomics data is high quality. They quantified 7,564 proteins across all models as well as 39,846 phosphopeptides corresponding to 29,311 phosphorylation sites on 3,791 proteins. But, I don't know why there are more phosphopeptides than phosphosites. Am I missing something here?

2) Insulin resistance proteome: Overall 37% of the proteome appeared changing in all models suggesting significant proteome level regulation of insulin resistance including 1,693 differentially expressed proteins in CI, 1,621 in DEX and 1,019 in TNF but only 12 proteins in MPQ and AA. Obviously there will be a question why there are so few proteins changing in MPQ and AA models? To my understanding mitochondrial oxidants generated by MPQ and AA are likely to have major impact on oxidative stress responses. Also, number of differentially regulated proteins in MPQ and AA do not match. Authors mentioned 12 (nine in MPQ, nine in AA and 6 in both) but that's 15 not 12.

3) Moreover, of the 1,951 insulin-response phosphopeptides, 767 were not regulated in any insulin resistance model. Only 128 were defective in three or more models and just 7 in all models. This is interesting and authors argued that insults causing insulin resistance do not just attenuate insulin signaling, rather they require insulin-responsive signaling networks. However, looking into the processes of these phosphopeptides, they are very broad (endocytosis, focal adhesion, transport, and more). While authors argue that such a poor overlap justifies studying multiple models but for me it seems opposite. I will assume a greater overlap among insulin-responsive phosphopeptides or phosphosites not matter what model we use. It would be important and interesting to evaluate at least

a few of the insulin-defective phosphosites for validation or deeper mechanistic understanding using point mutation or gene knockdown studies.

4) In pharmacological insulin resistance treatments (line 400), 0.5 and 100 nM insulin was used in wild type or HA-GLUT4-mRuby-3-expressing cells. Not sure why such a huge concentration difference was needed for these two treatments. On the other hand, for animal model 0.5 and 10 nM concentrations were used.

Finally, a paper which relies so heavily on phosphopeptide measurements, it would be really important to validate some of the results using complementary techniques. I think it's not overstatement to say that there are no boundaries of the MS technologies. Therefore, without some follow-up mechanistic studies either by gene KD or point mutation, enhanced understanding of underlying mechanisms of insulin signaling and development of insulin resistance would be limited and may preclude from its publication in such a high impact journal.

Reviewer #2 (Remarks to the Author):

In this report, Humphrey and colleagues carry out heroic studies designed to address the molecular events linked to insulin resistance with a focused effort targeting the phosphoproteome of adipocytes. The hypothesis is that by assessing different molecular mechanisms to induce insulin resistance, commonalities will be gleaned that point towards mechanisms and importantly, potential targets. As expected, the authors find defects in phosphoproteome that track with both basal, and insulin-stimulated signaling components. Like all experimental plans, the details matter and the authors have done a good job presenting their rationale for study and the workflow that pointed them towards GS3K signaling.

During review, the following questions arose:

1. The authors have chosen only a single time point to assess insulin action, 10 minutes. Have they done any characterization of this time point vis a vis others to determine if this is optimal?
2. The pharmacologic inhibition of GSK3 was done with a single molecule as opposed to a series of molecules to negate or diminish off target effects. Can this point be addressed experimentally, were other small molecules attempted?
3. Table 1 is presented very poorly and should be redrawn.
4. Figures 2d and 3d add little value and can be moved to supplement.
5. Only male mice were evaluated in this study. The rationale for this is that male mice develop insulin resistance more avidly than do female mice. That begs the question if the authors have assessed, even preliminarily, if GSK3 activity in females on a high fat diet is downregulated? This would add to the paper if such a correlate could be established.

Reviewer #3 (Remarks to the Author):

This manuscript applies a high throughput phosphoproteomics approach to assess common changes in various models of insulin resistance in adipocytes (both cell line and tissue models). The authors conclude that insulin resistance is associated with significant re-wiring of the pathways activated by insulin. Further analysis identified GSK3 (more precisely, failure to inhibit GSK3) as being a common element of insulin resistance. The authors have provided a significant amount of data (and supplemental data) and while focused on mass spectrometry, also include analysis of GLUT4 translocation as a read-out of downstream insulin signaling.

A strength of the study is the inclusion of multiple models for insulin resistance. As the authors note,

this also leads to greater biological complexity but this is poorly accounted for in data interpretation. For example, each of the “insults” may result in reduced insulin sensitivity but also have many other biological effects that are not common and may have indirect impact on insulin signaling. Homing in on the small amount of common changes does not necessarily account for other contributions to desensitization of response to insulin. The repeated use of the term “rewiring” is also not explained or justified. These effects are largely reversible and dynamic.

While this manuscript certainly represents significant work and the mass spectrometry data is technically well acquired, the biological interpretation of the data is somewhat over-simplified to focus on only one of the many mechanisms that are taking place and then pursues a single mechanism. Notably, the central conclusion is that the resistance insults similarly interfere with phosphorylation and inhibition of GSK3. However, knock-in alleles of the two inhibitory phosphorylation sites (Serines 21 and 9 in the alpha and beta isoforms) were not reported as being insulin resistant, nor did they show signs of diabetes (PMC1142569). These mice or cells with similar mutations would be strong tests of the conclusions made regarding GSK3.

This is a mass spectrometry based study but the authors conflate phosphorylation with activity (especially while inferring functional impact on GSK3). However, stoichiometry and actual effect on activity is neither directly measured nor reported. As the authors know, there are both compensatory and aggravating effects dependent on the balance/flux of phosphorylation at any given residue. The authors have measured a series of putative GSK3 targets and the impact on their phosphorylation and so may be able to provide some estimation of degree of inhibition (recognizing the substrate phosphorylation state is also a composite of kinase/phosphatase activity). This would strengthen the analysis.

Recognizing it may be disrespectful to ask, given quantitative mass spectrometry data in figure 2b, representative immunoblots of the phosphorylation levels of Serine 21/9 under the various states would allow visualization of the relative changes to GSK3 (while there are caveats to immunoblotting, at least the degree of loss of signal is a good indicator). Likewise for the data implying reduction in GSK3 protein levels. It would also be useful to indicate the relative level of tyrosine phosphorylated GSK3 peptide.

Minor comments:

Why was DIA used to acquire some of the phosphoproteomic data?

There is a lack of description of the tissue preparation for mass spec in methods. A comment on the appropriateness of the protocol and an indication of an awareness of the changes at the phosphoproteome level when excising tissues would also be useful.

A justification for the cut-offs chosen in the volcano plots should be provided (figure 4c).

Line 57: since both references calling the hypothesis into question are from one of the authors, this should be stated as such.

Table 1 is of questionable value as it has missing data and simplified/non-quantified directionality.

It may be my ignorance/convention, but the use of the # symbol in some of the figures is confusing and not well described in the legends.

On line 297, the authors state a sequence consensus for GSK3 substrates was pS/TXXXpS/T but in fact the authors appear to have used a less stringent consensus where the intervening X amino acids are 3-5 in number which better reflects known targets. The looser consensus should be shown in the main text.

Reviewer #1 (Remarks to the Author):

However, at the end, my enthusiasm was dampened by the lack of validation or follow-up studies to obtain better biological insights that may preclude its publication in Nature Communication.

We appreciate the overall positivity of the reviewer. In an effort to address the Reviewer's concern about lack of validation, as described in detail below, we have now performed additional validation of GSK3 and validation of a completely new target of insulin signalling. We believe this significantly strengthens the manuscript.

Comments:

1) Their phosphoproteomics data is high quality. They quantified 7,564 proteins across all models as well as 39,846 phosphopeptides corresponding to 29,311 phosphorylation sites on 3,791 proteins. But, I don't know why there are more phosphopeptides than phosphosites. Am I missing something here?

One phosphosite can be present on multiple unique phosphopeptides. For example, the phosphosite pS1036 on Map3k5 is found on a phosphopeptide containing just pS1036, as well as on a phosphopeptide containing both pS1036 and the nearby site pS1040. This means there are more unique phosphopeptides than unique phosphosites.

2) Insulin resistance proteome: Overall 37% of the proteome appeared changing in all models suggesting significant proteome level regulation of insulin resistance including 1,693 differentially expressed proteins in CI, 1,621 in DEX and 1,019 in TNF but only 12 proteins in MPQ and AA. Obviously there will be a question why there are so few proteins changing in MPQ and AA models? To my understanding mitochondrial oxidants generated by MPQ and AA are likely to have major impact on oxidative stress responses.

Cells were chronically exposed to CI, DEX, and TNF for at least 1 d and up to 8 d for DEX, allowing sufficient time for the proteome to change. By contrast, cells were only exposed to MPQ or AA for 2 h. As we have shown previously in the context of stem cell differentiation (Yang et al., 2019), this is insufficient time for substantial protein abundance changes to occur. We have now added the following text to explain this in the Results subsection "Characterization of the insulin-resistant proteome":

"In contrast, only 9 proteins were differentially regulated in response to MPQ and 9 in response to AA (6 common to both). The absence of widespread proteome changes in these conditions reflects the acute treatment duration (2 h), since in contrast to the phosphoproteome the proteome requires between 3-6 hours to enact dynamic expression changes."

Also, number of differentially regulated proteins in MPQ and AA do not match. Authors mentioned 12 (nine in MPQ, nine in AA and 6 in both) but that's 15 not 12.

In total there were 12 proteins changed in either MPQ or AA. 6 were changed in both MPQ and AA, 3 were changed exclusively in MPQ, and 3 were changed exclusively in AA, making 12. This means that in total, 9 were changed in MPQ and 9 were changed in AA. We have modified the text to make this clearer in the Results subsection "Characterization of the insulin-resistant proteome":

"In contrast, only 9 proteins were differentially regulated in response to MPQ and 9 in response to AA (6 common to both)"

3) Moreover, of the 1,951 insulin-response phosphopeptides, 767 were not regulated in any insulin resistance model. Only 128 were defective in three or more models and just 7 in all models. This is interesting and authors argued that insults causing insulin resistance do not just attenuate insulin signaling, rather they require insulin-responsive signaling networks. However, looking into the

processes of these phosphopeptides, they are very broad (endocytosis, focal adhesion, transport, and more). While authors argue that such a poor overlap justifies studying multiple models but for me it seems opposite. I will assume a greater overlap among insulin-responsive phosphopeptides or phosphosites not matter what model we use. It would be important and interesting to evaluate at least a few of the insulin-defective phosphosites for validation or deeper mechanistic understanding using point mutation or gene knockdown studies.

We suggest that the reviewer may have misinterpreted the passage in question by conflating insulin-responsive phosphopeptides with defective phosphopeptides. Of the 1,951 phosphopeptides that respond to insulin in CTRL cells, 767 are not regulated by insulin (defective) in one or more models of insulin resistance. It is not the case that the overlap between insulin-regulated phosphopeptides across models is low. The opposite is true - the overlap between defects in insulin signalling (sites that lose their response to insulin) across models is low. This suggests that each model of insulin resistance induces unique changes to the insulin signalling network.

We then focussed on insulin-regulated phosphopeptides that were defective across multiple models, on the assumption that these would be more likely to contribute to insulin action and insulin resistance. This led us to probe the kinases MARK2 and MARK3, which contain insulin-regulated phosphopeptides defective across 4 insulin resistance models, by siRNA-mediated knockdown and pharmacological inhibition. These experiments established a novel role for the kinases MARK2 and MARK3 in insulin-stimulated GLUT4 translocation. We believe this addresses the reviewer's concern about evaluating insulin-defective phosphosites, and highlights that focussing on insulin signalling impaired across multiple insulin resistance models can reveal novel regulators of insulin-stimulated GLUT4 translocation.

MARK2/3 data are in Fig. 2E-F and S4E-G, and are described in the Results subsection "Characterizing defective and emergent insulin signaling":

"Finally, we rationalized that proteins with phosphosites defective across most models would be likely to contribute to insulin action and resistance. The 37 phosphopeptides defective in four or more models included S453 and S469 on the kinases MARK2/3 (**Fig. 2E**). These phosphosites correlated positively with a subset of MARK2/3 substrates (**Fig. S4E**) and have high phosphosite functionality scores as established by Ochoa et al.¹ (0.68 and 0.58) indicating they may enhance MARK2/3 activity. As these sites were decreased by insulin in CTRL cells, we hypothesized that MARK2/3 may negatively regulate GLUT4 translocation. Consistent with this, siRNA-mediated knockdown of *MARK2/3* (approximately 50% reduction **Fig. S4F**) or pharmacological inhibition of MARK2/3 (confirmed by reduced phosphorylation of the substrate HDAC **Fig. S4G**) increased insulin-stimulated GLUT4 translocation (**Fig. 2F-G**). These data agree with the adipose insulin hypersensitivity observed in MARK2 knock-out mice², and highlight S453/S469 as potential regulatory nodes through which MARK2/3 may contribute to insulin action."

4) In pharmacological insulin resistance treatments (line 400), 0.5 and 100 nM insulin was used in wild type or HA-GLUT4-mRuby-3-expressing cells. Not sure why such a huge concentration difference was needed for these two treatments. On the other hand, for animal model 0.5 and 10 nM concentrations were used.

0.5 nM insulin was used as a physiological dose of insulin. This dose corresponds to the half maximal concentration of insulin required to stimulate GLUT4 translocation. Maximum stimulation is achieved at doses >~5 nM. Thus, while we acknowledge that 10 and 100 nM doses are very high they were simply used to invoke maximum stimulation. In our studies of insulin action over the years we have never observed a significant difference in response between 10 and 100 nM.

Finally, a paper which relies so heavily on phosphopeptide measurements, it would be really important to validate some of the results using complementary techniques. I think it's not

overstatement to say that there are no boundaries of the MS technologies. Therefore, without some follow-up mechanistic studies either by gene KD or point mutation, enhanced understanding of underlying mechanisms of insulin signaling and development of insulin resistance would be limited and may preclude from its publication in such a high impact journal.

As described above, we have now leveraged our analysis of phosphosites dysregulated across multiple models to prioritise new regulators of insulin action, which included the kinases MARK2 and MARK3. Following the reviewer's suggestion, we have used siRNA-based gene knockdown to study the roles of these proteins in GLUT4 translocation, which we augmented with experiments employing small molecule inhibition. These orthogonal approaches both demonstrated that lowering MARK2/3 activity increases insulin-stimulated GLUT4 translocation, implicating MARK2/3 as negative regulators of GLUT4 trafficking. MARK2/3 data are in Fig. 2E-F and S4E-G, and are described in the Results subsection "Characterizing defective and emergent insulin signaling" as provided above.

Our phosphoproteomics represents the most in-depth characterization of insulin signalling in insulin resistant adipocytes/adipose tissue to date. The novel experiments highlighted above, along with our comprehensive studies of GSK3, highlight the utility of our phosphoproteomics data for identifying novel regulators of insulin action and resistance.

Reviewer #2 (Remarks to the Author):

In this report, Humphrey and colleagues carry out heroic studies designed to address the molecular events linked to insulin resistance with a focused effort targeting the phosphoproteome of adipocytes. The hypothesis is that by assessing different molecular mechanisms to induce insulin resistance, commonalities will be gleaned that point towards mechanisms and importantly, potential targets. As expected, the authors find defects in phosphoproteome that track with both basal, and insulin-stimulated signaling components. Like all experimental plans, the details matter and the authors have done a good job presenting their rationale for study and the workflow that pointed them towards GS3K signaling.

During review, the following questions arose:

1. The authors have chosen only a single time point to assess insulin action, 10 minutes. Have they done any characterization of this time point vis a vis others to determine if this is optimal?

The main phosphoproteomics data were generated in 3T3-L1 adipocytes. We have previously characterised the dynamics of insulin signalling in this cell line using a time course from 15 sec to 60 min (Humphrey et al., 2013). We found that at 10 min, most insulin-responsive phosphosites had already undergone their maximum response without reverting to baseline. Hence, using this timepoint would give a comprehensive view of insulin signalling and its changes in insulin resistance. Furthermore, GLUT4 translocation and glucose uptake are promoted within the first few minutes of insulin stimulation (Tan et al., 2012). Phosphosites that respond to insulin after 10 min are unlikely to be relevant to glucose uptake and thus do not inform on insulin resistance. For these reasons we chose 10 min. We used 10 min insulin-stimulation in the *in vivo* mouse phosphoproteomes to match the 3T3-L1 data.

We have addressed this and other limitations of our study with a new section following the discussion called “Limitations of study”.

“Our studies were restricted to a single time point post insulin-stimulation (10 min.) – we reasoned that signaling events most relevant to glucose uptake should occur by this time since insulin stimulates GLUT4 translocation within 10 minutes⁷³ and most acute insulin-induced phosphorylation changes also occur within this timeframe¹⁸. We therefore cannot exclude the possibility that insulin resistance involves impaired signaling kinetics. “

2. The pharmacologic inhibition of GSK3 was done with a single molecule as opposed to a series of molecules to negate or diminish off target effects. Can this point be addressed experimentally, were other small molecules attempted?

We presented data using two distinct GSK3 inhibitors in HA-GLUT4-mR cells in the initial submission (Initial figure Fig. 5A-D). Both inhibitors improved insulin responses in the TNF and DEX cell culture models. We have extended these studies to now include a third GSK3 inhibitor in both HA-GLUT4-mR cells and WT cells (revised figures Fig. 5A-D, Fig. S7A-E). These studies show that three distinct inhibitors of GSK3 improve insulin-stimulated GLUT4 translocation in the TNF and DEX models of insulin resistance. The use of three distinct inhibitors and two models of GLUT4 translocation substantially minimises the chance that the effect we have observed is due to off-target effects.

These data are presented in Fig. 5A-D and S7A-E, and are described in the Results subsection “Pharmacological inhibition of GSK3 rescues insulin sensitivity *in vitro* and *ex vivo*”:

“In our analysis of 3T3-L1 models of adipocyte insulin resistance we detected multiple insulin signaling alterations shared across different models, leading us to hypothesize that pharmacological targeting of

any one of these alterations may only confer a partial reversal of insulin resistance. As impaired deactivation of GSK3 by insulin was observed in all insulin resistant 3T3-L1 models and insulin resistant adipose tissue, we decided to test this hypothesis on GSK3. We treated insulin-resistant 3T3-L1 adipocytes (CI, DEX, and TNF) with the GSK3 inhibitors GSK3i, CHIR99021 (CHIR), or AZD2858 (AZD) 1.5 h prior to administration of insulin, and monitored plasma membrane GLUT4 using an antibody that recognizes an exofacial region of GLUT4⁵, or by stably expressing an HA-GLUT4-mRuby3 reporter construct. GSK3 inhibition was confirmed by reduced phosphorylation of glycogen synthase (**Fig. S7A**). All three insulin resistance treatments impaired insulin-stimulated GLUT4 translocation in response to 0.5 or 100 nM insulin in wild type or HA-GLUT4-mRuby3-expressing cells (**Fig. 5A, 5C, S7B-E**). GSK3 inhibition increased cell surface endogenous GLUT4 and HA-GLUT4-mRuby3 in control, TNF and DEX-treated cells. The response to GSK3 inhibition was much greater in TNF and DEX models than in control cells (**Fig. 5B, D**) so that in general insulin responses were nearly equivalent to control cells. However, CI-treated cells were largely refractory to GSK3 inhibition (**Fig. 5A-D, S7B-E**). Notably, the effects of these GSK3 inhibitors in the DEX and TNF models were only observed in insulin-stimulated cells, suggesting that this intervention acts by specifically augmenting the insulin response.”

3. Table 1 is presented very poorly and should be redrawn.

Table 1 has now been removed from the manuscript.

4. Figures 2d and 3d add little value and can be moved to supplement.

We have moved these figures to supplement. Figure 2D is now S4C and Figure 3D is now S5C.

5. Only male mice were evaluated in this study. The rationale for this is that male mice develop insulin resistance more avidly than do female mice. That begs the question if the authors have assessed, even preliminarily, if GSK3 activity in females on a high fat diet is downregulated? This would add to the paper if such a correlate could be established.

As the reviewer points out, the work presented in our manuscript is limited to adipose tissue from male mice. We have historically used male mice and developed our acute time course of insulin resistance and reversal of insulin resistance in male mice. We agree with the reviewer that it will be important to know whether the observations we have made on signalling changes in insulin resistance, including increased GSK3 activity (and partial reversal of insulin resistance with GSK3 inhibition) is also true in females. However, since we do not see simple correlations between GSK3 phosphorylation and GSK3 activity in insulin resistance, it is not trivial for us to assess GSK3 activity in females, as this would require another large-scale phosphoproteomics study.

We have addressed this and other limitations of our study with a new section following the discussion called “Limitations of study”:

“Our studies were undertaken exclusively in male mice, and future efforts should establish whether the signaling changes observed also occur in female mice.”

Reviewer #3 (Remarks to the Author):

A strength of the study is the inclusion of multiple models for insulin resistance. As the authors note, this also leads to greater biological complexity but this is poorly accounted for in data interpretation. For example, each of the “insults” may result in reduced insulin sensitivity but also have many other biological effects that are not common and may have indirect impact on insulin signaling. Homing in on the small amount of common changes does not necessarily account for other contributions to desensitization of response to insulin.

We thank the reviewer for highlighting the strength of including multiple models of insulin resistance. As we have previously noted (Fazakerley et al., 2018; Hoehn et al., 2009), searching for molecular changes between these models, which have the same phenotype of impaired insulin responses, is a powerful method for identifying causes of insulin resistance. We acknowledge that this method is not comprehensive in that each model will result in unique changes in protein levels and phosphorylation that could play a role in causing insulin resistance. However, these changes will be much harder to identify, as it is challenging to separate them from the intrinsic effects of each insult that may not contribute to insulin resistance. Finally, while we have focussed our analysis on signalling changes shared across multiple models of insulin resistance, we have strived to make our data readily accessible through a user-friendly web app and in supplemental tables. This will allow other researchers to interrogate different facets of our data, such as model-specific (phospho)proteomic alterations.

We have addressed this and other limitations of our study with a new section following the discussion called “Limitations of study”:

“Finally, we have focused largely on proteomic and phosphoproteomic changes occurring across multiple cultured and in vivo models of insulin resistance as we have previously found this approach is successful in identifying causal drivers of metabolic dysfunction. However, this does not account for model-specific changes that contribute to insulin resistance. Bona fide changes of this type may contribute meaningfully to insulin resistance within each model, but they are challenging to identify as they cannot be readily separated from changes unrelated to insulin resistance.”

The repeated use of the term “rewiring” is also not explained or justified. These effects are largely reversible and dynamic.

We thank the reviewer for flagging this issue of clarity. We now explicitly define “rewiring” in the Results subsection “Insulin resistance rewires the insulin signaling network”:

“Herein we define “rewiring” as the selective, potentially reversible weakening or strengthening of multiple connections in the insulin signaling network.”

This definition is appropriate as we observe hundreds of insulin-regulated phosphorylation events that are altered in insulin resistance, some being impaired and others enhanced. As the reviewer points out, such changes are expected to be reversible and dynamic.

While this manuscript certainly represents significant work and the mass spectrometry data is technically well acquired, the biological interpretation of the data is somewhat over-simplified to focus on only one of the many mechanisms that are taking place and then pursues a single mechanism.

We completely agree with the reviewer on this point and indeed this is one of the major challenges of these kinds of systematic analyses. It is our view that global analyses of protein phosphorylation provide new insights into signaling networks. Studying one or two signaling proteins, as we have done here, highlights the utility of our approach to identify regulators of insulin action. But in the end,

such studies offer enormous resources for the research community and this has always been our philosophy. Indeed, many labs have used our previously published phosphoproteomes to discover new targets.

Furthermore, in our biological interpretation of the data we have attempted to highlight that there appear to be not just one or two, but many unrelated signalling changes contributing to insulin resistance. This view of insulin resistance is distinct from one we see as commonly held in the field, namely that insulin resistance is due to a single defect in a proximal insulin signalling component such as Akt or IRS1/2. This is an important message from our manuscript, as such we attempted to capture this central theme in the title of our manuscript and at the end of the Results subsection “Characterizing defective and emergent insulin signaling”:

“In all, rewiring of the insulin signaling network was largely exclusive of well-studied insulin signaling proteins. Instead, changes observed comprised a set of kinases, proteins, and pathways without known links with insulin at present. This suggests our knowledge of the extent of mechanistic insulin signaling remains far from complete. Our data also imply that multiple parallel signaling alterations may cumulatively contribute to insulin resistance. If this is the case, targeting a single kinase or pathway may only confer a partial benefit to insulin sensitivity.”

We believe that our work on GSK3 beautifully addresses this concept as when we disrupt GSK3 this only partially rescues insulin resistance. We interpret this to mean that insulin resistance represents many independent defects and this has important ramifications for therapeutic development. We draw the reviewer’s attention to the first paragraph of the Discussion where we have attempted to make this stance clear: ’

“By utilizing standardized and controlled cell culture models spanning a wide range of known contributors to insulin resistance, we observed that insulin resistance does not involve a simple signaling defect, but instead a profound rewiring of many nodes within the insulin signaling network. Pharmacologically targeting one of these nodes in isolation improved insulin sensitivity, but only partially. This highlights the utility of our approach in identifying targets to ameliorate metabolic dysfunction, and also supports the concept that insulin resistance is a complex multi-nodal defect. Correction of this defect will either require identification of common regulators of these discrete nodes or treatments designed to target multiple nodes.”

Notably, the central conclusion is that the resistance insults similarly interfere with phosphorylation and inhibition of GSK3. However, knock-in alleles of the two inhibitory phosphorylation sites (Serines 21 and 9 in the alpha and beta isoforms) were not reported as being insulin resistant, nor did they show signs of diabetes (PMC1142569). These mice or cells with similar mutations would be strong tests of the conclusions made regarding GSK3.

We would like to point out that the central conclusion regarding GSK3 is that insulin resistance interferes with inhibition of GSK3 by insulin, however this is not entirely through altered phosphorylation of GSK3a/b at S21/S9. Notably, phosphorylation of these sites is intact in the MPQ and AA models despite impaired GSK3 inhibition as detected by KSEA (Fig. 2A-B).

We acknowledge there is a discrepancy between our finding that acute inhibition of GSK3 can improve insulin action during insulin resistance, and the lack of an insulin resistant phenotype in GSK3 knock-in mice (McManus et al., 2005). This discrepancy is not unique to our study, as it has been noted before that in most cases acute GSK3 inhibition has affected glucose metabolism whereas genetic manipulation has not (MacAulay & Woodgett, 2008). We suggest acute inhibition may be more appropriate in the context of acute insulin action. In particular, phosphorylation signaling networks are highly adaptable to chronic perturbations especially those involving germ line mutations. As a salient example, mice with muscle-specific deletion of Akt1 and Akt2 are unable to activate Akt signalling in response to insulin, yet they retain normal levels of insulin-stimulated glucose uptake through a compensatory mechanism dependent on AMPK (Jaiswal et al., 2019). Furthermore, the

authors of the paper reporting S21/S9 knock-in alleles (McManus et al., 2005) acknowledge that they "are unable to rule out that the constitutive knock-in of GSK3 isoforms leads to compensatory mechanisms during development, resulting in an absence of a diabetic phenotype". To further bolster our findings we have now used three independent inhibitors of GSK3 and achieved similar results (Fig. 5A-D, S7A-E). In light of these results and the caveats discussed above, we believe we have sufficiently defended our conclusions regarding GSK3 in insulin resistance.

We have addressed the discrepancy between pharmacological inhibition and genetic activation of GSK3 in the 2nd paragraph of the Discussion:

“In this study we acutely administered GSK3 inhibitors to restore the acute inhibition of GSK3 normally achieved by insulin, which partially restored insulin sensitivity. However, studies in GSK3 α/β S21/S9A knock-in mice – which are refractory to insulin inhibition to begin with – report normal muscle insulin-stimulated glucose transport⁸. This disparity between pharmacological inhibition and genetic activation of GSK3 may be explained by the different tissues studied (adipose vs skeletal muscle), or compensatory mechanisms invoked by the constitutive knock-in of the S21/S9A mutation. Knowledge of how GSK3 is dysregulated in insulin resistance could resolve this disparity.”

This is a mass spectrometry based study but the authors conflate phosphorylation with activity (especially while inferring functional impact on GSK3). However, stoichiometry and actual effect on activity is neither directly measured nor reported. As the authors know, there are both compensatory and aggravating effects dependent on the balance/flux of phosphorylation at any given residue. The authors have measured a series of putative GSK3 targets and the impact on their phosphorylation and so may be able to provide some estimation of degree of inhibition (recognizing the substrate phosphorylation state is also a composite of kinase/phosphatase activity). This would strengthen the analysis.

We believe that the most meaningful way to define the *in vivo* activity of a kinase (*in vivo* meaning in live tissue or cells) is through the phosphorylation status of a comprehensive list of kinase targets, as we have achieved using KSEA. After all, the biological effect of kinase activation is achieved through phosphorylation of its substrates. We have clarified this definition of kinase activity in the manuscript in the Results subsection “Profiling kinase regulation in insulin resistance:

“To identify potential kinases involved, we performed kinase substrate enrichment analysis (KSEA)⁹, which assessed changes in *in vivo* effective kinase activity across the insulin resistance models using the insulin response of previously annotated kinase substrates (PhosphositePlus¹⁰, Fig. 2A).”

Measurements of intrinsic kinase activity, such as those obtained by an *in vitro* kinase assay, are likely not as meaningful because they generally do not capture crucial *in vivo* context such as a kinase’s protein-protein interactions or localisation, which can be key determinants of substrate phosphorylation. We recognise a caveat to this method is that some substrates of a kinase will also be targeted by phosphatases and additional kinases. However, we believe this is unlikely to substantially affect estimates of GSK3 kinase activity, for instance, given the large number of GSK3 substrates we identify and use for KSEA. The issue of stoichiometry raised by the referee is also interesting as it represents a key reason for assessing kinase activity as we do here. We have previously shown that kinases like Akt only require very low stoichiometry of phosphorylation at their activating sites in order to achieve maximal substrate phosphorylation, and hence maximal *in vivo* activity (Tan et al., 2012). Thus, measurements of *in vitro* Akt activity or the stoichiometry of Akt regulatory sites provide limited information as they do not necessarily correlate with the phosphorylation of Akt’s substrates. Finally, we believe our KSEA analysis provides exactly what the reviewer has asked for - it is a measure of the degree of GSK3 inhibition based on the phosphorylation of putative GSK3 targets.

Recognizing it may be disrespectful to ask, given quantitative mass spectrometry data in figure 2b, representative immunoblots of the phosphorylation levels of Serine 21/9 under the various states would allow visualization of the relative changes to GSK3 (while there are caveats to immunoblotting, at least the degree of loss of signal is a good indicator). Likewise for the data implying reduction in GSK3 protein levels. It would also be useful to indicate the relative level of tyrosine phosphorylated GSK3 peptide.

Representative immunoblots of pS21/S9 Gsk3a/b and total Gsk3b from two biological replicates have been included in Fig. S5E. The level of phosphorylated Gsk3a Y279 has been included in Fig. S5E and Fig. S6H. The Y279 levels in 3T3-L1 adipocyte data have been interpreted in the Results subsection “Insulin-regulated inhibition of the kinase GSK3 is impaired in insulin resistance”:

“Phosphorylation of GSK3 α/β at Y279/Y216 has been reported to enhance kinase activity and may be an autophosphorylation event. We identified no significant insulin response in GSK3 α Y279, and GSK3 β Y216 was not quantified (**Fig. S5G**). However, GSK3 α Y279 was weakly decreased by insulin in CTRL cells but not decreased in DEX, TNF, and AA. Although we cannot exclude the involvement of tyrosine phosphorylation in impaired deactivation of GSK3, the small magnitude of Y279 changes between models suggests that any contribution to dysregulated GSK3 function may be minor.”

The Y279 levels in adipose tissue data have been interpreted in the subsection “Signaling rewiring and GSK3 dysregulation in insulin resistant adipose tissue”:

“Phosphorylation of S21 on GSK3 α was not adequately quantified, and phosphorylation of Y279 on GSK3 α was not changed by insulin (**Fig. S6H**).”

Y216 on Gsk3b was not quantified in phosphoproteomics data.

Minor comments:

Why was DIA used to acquire some of the phosphoproteomic data?

The 3T3-L1 insulin resistance phosphoproteome was initially acquired using DDA. Given the centrality of this dataset to the paper, once DIA phosphoproteomics methods became widespread (particularly software tools for the analysis of the data) the data were re-acquired by DIA. The remaining two phosphoproteomics datasets were not re-acquired by DIA as the DDA data already sufficiently supported the main findings of the project.

There is a lack of description of the tissue preparation for mass spec in methods. A comment on the appropriateness of the protocol and an indication of an awareness of the changes at the phosphoproteome level when excising tissues would also be useful.

We have now added additional detail to the Methods section describing how tissues were kept cold during processing to avoid changes to the phosphoproteome.

A justification for the cut-offs chosen in the volcano plots should be provided (figure 4c).

We have now clarified that the cutoffs were chosen to identify phosphopeptides increased or decreased by GSK3 inhibition by 1.5-fold or greater. Along with the presence of a GSK3 substrate motif, we consider this sufficient to establish a site as a putative GSK3 substrate, since existing GSK3 substrates passed this filtering as highlighted in the Results subsection “Targeted analysis of the dephosphorylation defect in adipocytes”:

“The resulting 290 phosphopeptides corresponded to 274 phosphosites on 184 proteins (Fig. 3B, C, Data S5), and contained previously identified GSK3 targets such as S641, S645 and S649 on glycogen synthase¹², and S514 and S518 on DPYSL2^{13,14}, confirming the validity of our approach.”

Line 57: since both references calling the hypothesis into question are from one of the authors, this should be stated as such.

This has now been clarified in the manuscript:

“However, we and others have recently called this hypothesis into question¹⁵⁻¹⁷, and the causal relationship between changes in early insulin signaling and the downstream actions of insulin remains unclear, especially in the context of insulin resistance.”

Table 1 is of questionable value as it has missing data and simplified/non-quantified directionality.

Table 1 has now been removed from the manuscript.

It may be my ignorance/convention, but the use of the # symbol in some of the figures is confusing and not well described in the legends.

This has now been clarified in figures and legends.

On line 297, the authors state a sequence consensus for GSK3 substrates was pS/TXXXXpS/T but in fact the authors appear to have used a less stringent consensus where the intervening X amino acids are 3-5 in number which better reflects known targets. The looser consensus should be shown in the main text.

This has been corrected in Fig. 3A and on line 297:

“To enrich these data for sites more likely to be direct substrates of GSK3 in adipocytes (as opposed to downstream of GSK3), we selected only those that were down-regulated in response to GSK3i and also contained the motif of a GSK3 substrate (pS/T X(2-4) pS/T).”

References

- Fazakerley, D. J., Chaudhuri, R., Yang, P., Maghzal, G. J., Thomas, K. C., Krycer, J. R., Humphrey, S. J., Parker, B. L., Fisher-Wellman, K. H., Meoli, C. C., Hoffman, N. J., Diskin, C., Burchfield, J. G., Cowley, M. J., Kaplan, W., Modrusan, Z., Kolumam, G., Yang, J. Y. H., Chen, D. L., ... James, D. E. (2018). Mitochondrial CoQ deficiency is a common driver of mitochondrial oxidants and insulin resistance. *ELife*. <https://doi.org/10.7554/eLife.32111>
- Hoehn, K. L., Salmon, A. B., Hohnen-Behrens, C., Turner, N., Hoy, A. J., Maghzal, G. J., Stocker, R., van Remmen, H., Kraegen, E. W., Cooney, G. J., Richardson, A. R., & James, D. E. (2009). Insulin resistance is a cellular antioxidant defense mechanism. *Proceedings of the National Academy of Sciences of the United States of America*, 106(42). <https://doi.org/10.1073/pnas.0902380106>
- Humphrey, S. J., Yang, G., Yang, P., Fazakerley, D. J., Stöckli, J., Yang, J. Y., & James, D. E. (2013). Dynamic adipocyte phosphoproteome reveals that akt directly regulates mTORC2. *Cell Metabolism*. <https://doi.org/10.1016/j.cmet.2013.04.010>
- Jaiswal, N., Gavin, M. G., Quinn, W. J., Luongo, T. S., Gelfer, R. G., Baur, J. A., & Titchenell, P. M. (2019). The role of skeletal muscle Akt in the regulation of muscle mass and glucose homeostasis. *Molecular Metabolism*, 28. <https://doi.org/10.1016/j.molmet.2019.08.001>

- MacAulay, K., & Woodgett, J. R. (2008). Targeting glycogen synthase kinase-3 (GSK-3) in the treatment of Type 2 diabetes. In *Expert Opinion on Therapeutic Targets* (Vol. 12, Issue 10). <https://doi.org/10.1517/14728222.12.10.1265>
- McManus, E. J., Sakamoto, K., Armit, L. J., Ronaldson, L., Shpiro, N., Marquez, R., & Alessi, D. R. (2005). Role that phosphorylation of GSK3 plays in insulin and Wnt signalling defined by knockin analysis. *EMBO Journal*, *24*(8). <https://doi.org/10.1038/sj.emboj.7600633>
- Tan, S. X., Ng, Y., Meoli, C. C., Kumar, A., Khoo, P. S., Fazakerley, D. J., Junutula, J. R., Vali, S., James, D. E., & Stöckli, J. (2012). Amplification and demultiplexing in insulin-regulated Akt protein kinase pathway in adipocytes. *Journal of Biological Chemistry*, *287*(9). <https://doi.org/10.1074/jbc.M111.318238>

REVIEWERS' COMMENTS

Reviewer #1 (Remarks to the Author):

This manuscript represents a comprehensive proteomics and phosphoproteomics study of insulin resistance using different cell and animal models. Overall, experiments are well designed and executed and the manuscript is also well written. This is one of the most in-depth characterizations of insulin signaling pathways. One of the major concerns that I raised in my first review was the lack of validation or follow-up studies. Authors have now performed additional validation studies using siRNA based gene KD to study the roles of MAPK2/3 in GLUT4 translocation, and supports phosphoproteomics data. This, I believe, improves the quality and the significance of the study. I appreciate authors' efforts to address this and my other comments, and recommend this manuscript for publication in Nature Communication.

Reviewer #2 (Remarks to the Author):

In this revised manuscript, Humphrey and colleagues have addressed each of my major concerns adequately. They have added a section in the Discussion on "Limitations of the Study" that includes considerations such as kinetics, sex of animals, etc. that allows for consideration of caveats. Based on the sum of reviewers 1-3 comments, several new experiments that strengthen the manuscript have been added and some relatively unimportant considerations (original Table 1) have been deleted or moved to the Supplementary materials. The main conclusion based on a combination of phosphoproteomics, RNAi and inhibitor studies is that GSK3 is a/the major dysregulated node in insulin resistance. This is significant, highly unappreciated and introduces novel therapeutic opportunities not currently employed.

Reviewer #3 (Remarks to the Author):

The authors have responded constructively to most of the points raised (reviewer 3) and clarified some of the issues regarding the methodology. In particular, caveats of some of the interpretations of the dataset are included. There remain some concerns about the global effects of some of the insulin resistance inducers but this is mentioned in the limitations section. It would be informative to understand how insulin impacts GSK3 outside of the N-terminal phosphorylation sites and this discrepancy with the genetic analysis is somewhat dismissed as a compensatory effect - if so, does that still involve GSK3 or is it another pathway(s)? It would also be reassuring to see at least some orthogonal immunoblots in the main figures (e.g. figure 2) rather than relegated to the supplementary figures but will leave that to the editorial office.

This reviewer finds the overall contribution to the issue of insulin resistance mechanisms is significant and there is always a temptation to focus interpretation of such a dataset on a specific pathway (the opposite is also true, that lack of focus can be critiqued). Hence, the revised manuscript is both noteworthy and contributes to the field.

Reviewer #1 (Remarks to the Author):

This manuscript represents a comprehensive proteomics and phosphoproteomics study of insulin resistance using different cell and animal models. Overall, experiments are well designed and executed and the manuscript is also well written. This is one of the most in-depth characterizations of insulin signaling pathways. One of the major concerns that I raised in my first review was the lack of validation or follow-up studies. Authors have now performed additional validation studies using siRNA based gene KD to study the roles of MAPK2/3 in GLUT4 translocation, and supports phosphoproteomics data. This, I believe, improves the quality and the significance of the study. I appreciate authors' efforts to address this and my other comments, and recommend this manuscript for publication in Nature Communication.

We thank the Reviewer for their comments and for their time and effort in reviewing our manuscript. With their feedback we believe the revisions made have significantly strengthened the manuscript.

Reviewer #2 (Remarks to the Author):

In this revised manuscript, Humphrey and colleagues have addressed each of my major concerns adequately. They have added a section in the Discussion on "Limitations of the Study" that includes considerations such as kinetics, sex of animals, etc. that allows for consideration of caveats. Based on the sum of reviewers 1-3 comments, several new experiments that strengthen the manuscript have been added and some relatively unimportant considerations (original Table 1) have been deleted or moved to the Supplementary materials. The main conclusion based on a combination of phosphoproteomics, RNAi and inhibitor studies is that GSK3 is a/the major dysregulated node in insulin resistance. This is significant, highly unappreciated and introduces novel therapeutic opportunities not currently employed.

We thank the Reviewer for their comments and for highlighting strengths of the work. We appreciate their time and efforts, which have greatly helped to improve our manuscript.

Reviewer #3 (Remarks to the Author):

The authors have responded constructively to most of the points raised (reviewer 3) and clarified some of the issues regarding the methodology. In particular, caveats of some of the interpretations of the dataset are included. There remain some concerns about the global effects of some of the insulin resistance inducers but this is mentioned in the limitations section. It would be informative to understand how insulin impacts GSK3 outside of the N-terminal phosphorylation sites and this discrepancy with the genetic analysis is somewhat dismissed as a compensatory effect - if so, does that still involve GSK3 or is it another pathway(s)? It would also be reassuring to see at least some orthogonal immunoblots in the main figures (e.g. figure 2) rather than relegated to the supplementary figures but will leave that to the editorial office.

This reviewer finds the overall contribution to the issue of insulin resistance mechanisms is significant and there is always a temptation to focus interpretation of such a dataset on a specific pathway (the opposite is also true, that lack of focus can be critiqued). Hence, the revised manuscript is both noteworthy and contributes to the field.

We thank this Reviewer for highlighting these important points, which we have now addressed. We too will be interested to further understand the complex regulation of GSK3 phosphorylation, which we agree warrants future targeted studies. We have considered the reviewer's suggestions regarding immunoblots (i.e. Figure S5e -> Figure 2). Considering the global nature of the work, that these are the exact same biological samples that our phosphoproteomics/proteomics data were measured from, and the semi-quantitative nature of immunoblots in general, we feel these data are best situated as Supplementary Figures.